



# SUHMO: an AMR SUbglacial Hydrology MOdel v1.0

Anne M. Felden[1], Daniel F. Martin[1], and Esmond G. Ng[1]

[1]Applied Mathematics and Computational Research Division, Lawrence Berkeley National Laboratory, Berkeley, California, US

**Correspondence:** Anne M. Felden (amfelden@lbl.gov)

**Abstract.** Water flowing under ice sheets and glaciers can have a strong influence on ice dynamics, particularly through pressure changes, suggesting that a comprehensive ice sheet model should include the effect of basal hydrology. Modeling subglacial hydrology remains a challenge however, mainly due to the range of spatial and temporal scales involved - from subglacial channels to vast subglacial lakes. Additionally, networks of subglacial drainage channels dynamically evolve over time. To address some of these challenges, we have developed an Adaptive Mesh Refinement (AMR) model based on the Chombo software framework. We extend the model proposed by Sommers et al. (2018) with a few changes to accommodate the transition from unresolved to resolved flow features. We handle the strong nonlinearities present in the equations by resorting to an efficient nonlinear Full Approximation Scheme multigrid (FAS-MG) algorithm. We outline the details of the algorithm and present convergence analysis results demonstrating its effectiveness. Additionally, we present results validating our approach, using test cases from the Subglacial Hydrology Model Intercomparison Project (SHMIP) (de Fleurian et al., 2018). We finish by presenting a more complex AMR test case and discuss the effective pressure distribution as the spatial resolution increases.

## 1 Introduction

The extensive and accelerating retreat of glaciers observed over the last 150 years has fueled interest in the behavior of the cryosphere. It is common knowledge that ice sheets have grown and retreated many times over the past 2.6 million years, but some studies suggest that the current interglacial period is straying from its expected course and could last much longer than originally anticipated (Berger and Loutre, 2002; Ganopolski et al., 2016). The amount of water stored in the Greenland Ice Sheet (GrIS) alone has the potential to raise global-mean sea level (GMSL) by about 7 meters (Aschwanden et al., 2019), at a rate being characterized by deep uncertainties to external factors (Edwards et al., 2021). Glaciers and ice sheets contribution to GMSL have been increasing over the past decades, adding up to more than 50% of the total change over the period 2006-2018 (Edwards et al., 2021; Masson-Delmotte et al., 2021). Recent studies predict a *likely* GMSL rise of about 0.6 m by 2100 under intermediate greenhouse gas emissions scenarios, but ice sheet processes could drive GMSL to rise up to about 2 m by 2100 and 5 m by 2150 under a very high emissions scenario (Masson-Delmotte et al., 2021). Since more than 30% of the world's population today live in what can be considered coastal areas, predicting and understanding the mechanics governing the melting of the cryosphere can be considered a problem of pressing scientific and societal importance.

The evolution of the cryosphere can hardly be decoupled from that of its environment, and numerical ice sheet models should ideally include simplified but accurate representations of the interactions with the other components of the Earth system (see,





e.g., Fig.1 of Goelzer et al. (2017)). In particular, it is now widely accepted – through observational evidence and theoretical considerations – that basal hydrology strongly influences the dynamics of glaciers and ice sheets, mainly through changes in the basal pressure (e.g., Iken, 1981; Bindschadler, 1983; Fricker et al., 2007; Stearns et al., 2008; Doyle et al., 2018).

Concurrent ice uplift and acceleration have been observed both in the GrIS and in the Antarctic Ice Sheet (AIS) (Das et al., 2008; Nienow et al., 2017; Tuckett et al., 2019), suggesting significant routing of surface lakes to the base and an active and dynamic subglacial hydrology network. Persistent subglacial water structures like conduits – from one draining event to the next – and lakes have also been inferred in Antarctica, with potentially critical importance for the future of the AIS and GMSL projections (Siegfried and Fricker, 2018; Kirkham et al., 2019; Malczyk et al., 2020).

Modeling subglacial hydrology remains a challenge, mainly due to the large discrepancy in spatial and temporal scales between the physics driving subglacial phenomena versus those driving ice sheet dynamics. Relevant timescales for the latter are of the order of tens to thousands of years, when subglacial water cycles occur over days or months. Englacial structures can organize into channels a few meters wide, while Antarctica covers an area of over 10 million square kilometers. Additionally, the variety of subglacial and englacial water structures, including sheets and cavities, ice-walled conduits (Röthlisberger, 1972),

bedrock channels, lakes and everything in-between, make it difficult to construct a comprehensive subglacial hydrology model (see for example Fig. 2 of Flowers (2015)). To date, no complete, physically-based theory has been developed, and most models follow more or less the same "recipe" made up from drainage-system elements that are assembled depending on what is thought to be physically relevant and on the numerical framework at hand. These models typically resort to a differentiation between inefficient flow configurations (linked cavities) and efficient ones (channels). Both are described by a balance between

opening and closing terms: cavities typically open up due to sliding over the bedrock while channels open up due to dissipative-heating and melting of their walls, and closing is due to ice creep. While a review of all subglacial modeling efforts is out of the scope of the present paper, we briefly discuss recent relevant two-dimensional, multi-elements efforts having the ultimate goal of being coupled with an ice sheet model, to provide context to our study. The reader is referred to the thorough review of Flowers (2015), as well as to the references and participating models in the Subglacial Hydrology Model Intercomparison

Project (SHMIP) (de Fleurian et al., 2018) for additional information.

"Next-generation" efforts (terminology from Flowers (2015)) started emerging in the early 2000s, with pioneering work to simulate the Weichselian Scandinavian Ice Sheet by Arnold and Sharp (2002). In their study, ice velocities relied on calculations of subglacial water pressures and the use of a water pressure-dependent sliding law. The two dimensional basal hydrology is made up of either inefficient (linked cavities) or efficient (Röthlisberger channels) flow configurations – both cannot coexist

in the same cell, and findings highlighted that the spatial distribution of ice flow is greatly impacted by the presence of the subglacial water. More recently, mainly motivated by observations made in GrIS linking surface meltwater and basal sliding, the last decade has seen a sustained interest in comprehensive subglacial hydrology models. Schoof (2010) used a model with a single equation for the cross-sectional area of discrete conduits that could behave as either cavities or channels, to investigate the link between ice velocities and subglacial water channelization patterns generated by seasonal and short-term water supply

variations. Hewitt et al. (2012; 2013) introduced a two-dimensional subglacial hydrology model coupling a continuum sheet and discrete channel elements – each requiring a different set of equations. This model is linked to an ice flow model, enabling





a parametric study on a synthetic sheet-like geometry emulating GrIS margins. Werder et al. (2013) extended this effort to an unstructured finite element mesh where the distributed continuum sheet is solved using finite elements on a set of triangular cells while channels are located along the edges of the cells. The resulting model, the Glacier Drainage System model (GlaDS), is coupled to both the Ice-sheet and Sea-level System Model (ISSM) (Larour et al., 2012) and Elmer/Ice ice sheet model (Gagliardini et al., 2013) and has been successfully employed to study both the GrIS and AIS (e.g. Dow et al., 2016; Gagliardini and Werder, 2018). SHAKTI (Sommers et al., 2018), also coupled to ISSM, removes the distinction between channels and cavities/sheet by using a single set of equations to evolve the water gap height (unlike the model of Schoof (2010) that evolves a drainage cross-sectional area). Such an approach is attractive because water structures are free to evolve and merge anywhere in the domain (without being restricted to cell faces for example), without the need for an explicit coupling between them. It does, however, raise the question of how fine the mesh must be to properly resolve the various sublgacial features. The cost of finer resolution to accommodate the formation of channels could very quickly become prohibitive. Fortunately, fine structures are expected to only occur in very localized areas, making this a perfect target for Adaptive Mesh Refinement.

We follow the approach of Sommers et al. (2018) using the Chombo framework (Adams et al., 2001-2021) to implement an Adaptive Mesh Refinement (AMR) subglacial hydrology model. We propose a small modification to the set of equations presented in Sommers et al. (2018) in order to seamlessly transition from under-resolved to resolved channels, alleviating an unphysical asymptotic behavior as the mesh size begins to allow resolution of typical channel scales. Building on the AMR framework allows development of a robust numerical approach to solve the resulting nonlinear system of PDEs which achieves second order convergence in space.

The paper is structured as follows. In Sect. 2, we summarize the set of equations used in SUHMO. In Sect. 3 we provide the details of the nonlinear Full Approximation Scheme (FAS) algorithm employed to solve the governing equations of Sect. 2. Convergence analyses demonstrating the efficiency and accuracy of our implementation is presented in Sect. 4. We then present additional validating results, choosing three representative test cases extracted from SHMIP (de Fleurian et al., 2018), in Sect. 5. Results from a transient, larger scale, AMR simulation with random bed roughness and interesting topographic features are discussed in the final Sect. 6. We finish with concluding remarks.

## 2 Conservation equations for the subglacial drainage system

We start with a set of equations similar to that used in the SHAKTI model (Sommers et al., 2018), which is a parallelized, finite-element subglacial hydrology model currently implemented as part of the open-source Ice-sheet and Sea-level System Model (Larour et al., 2012). We will provide a brief overview, before introducing a novel diffusion component that we believe represents an improvement to the existing model. For additional details concerning the original equations, the reader is referred to Sommers et al. (2018).





### 2.1 Equations

The governing equation set starts with a two-dimensional expression for the conservation of mass – assuming we are dealing with an incompressible fluid:

$$\frac{\partial b}{\partial t} + \frac{\partial b_e}{\partial t} + \nabla \cdot \mathbf{q} = \frac{\dot{m}}{\rho_w} + e_s, \tag{1}$$

where $b$ is the subglacial water gap height (m), $b_e$ is the volume of water stored englacially per unit area of bed (m), $\mathbf{q}$ is the gap-integrated basal water flux (m$^2$ s$^{-1}$), $\dot{m}$ is the melt rate (kg m$^{-2}$s$^{-1}$), $\rho_w$ is the density of water (kg m$^{-3}$), and $e_s$ encompasses all external sources of meltwater (produced englacially or surface meltwater, for example) (m s$^{-1}$).

An approximate momentum equation for water velocity integrated over the gap height gives rise to an expression for the water flux, based on equations developed for flow in rock fractures (e.g., Zimmerman et al., 2004):

$$\mathbf{q} = \frac{-b^3 g}{12\nu(1 + \omega Re)} \nabla h, \tag{2}$$

where $g$ is the gravitational acceleration (m s$^{-2}$), $\nu$ is the kinematic viscosity of water (m$^2$ s$^{-1}$), $\omega$ is a dimensionless parameter controlling the nonlinear transition from laminar to turbulent flow and $Re$ is the Reynolds number. The hydraulic head $h$ (m) is defined:

$$h = \frac{P_w}{\rho_w g} + z_b, \tag{3}$$

where $P_w$ is the water pressure (Pa) and $z_b$ is the bed elevation (m). The Reynolds number follows a classical definition:

$$Re = \frac{|v|b}{\nu} = \frac{|\mathbf{q}|}{\nu}, \tag{4}$$

where $v$ is the average flow velocity across the gap height. Equation (2) is an important piece in the SHAKTI model (Sommers et al., 2018). It allows for a spatially and temporally variable hydraulic transmissivity in the system and facilitates the representation of the simultaneous coexistence of laminar, transitional, and turbulent flow in sub-regions of the domain.

The melt rate $\dot{m}$ includes heat produced at the bed (geothermal flux and frictional heat due to sliding over the bed) along with heat generated through internal dissipation (mechanical energy converted to thermal energy by the flow), which is effectively melting the drainage system's walls and ceiling:

$$\dot{m} = \frac{1}{L}(G + |\mathbf{u}_b \cdot \tau_b| - \rho_w g \mathbf{q} \cdot \nabla h + c_t c_w \rho_w \mathbf{q} \cdot \nabla P_w), \tag{5}$$

where $L$ is the latent heat of fusion of water (J kg$^{-1}$), $G$ is the geothermal flux (W m$^{-2}$), $\mathbf{u}_b$ is the ice basal velocity vector (m s$^{-1}$), $\tau_b$ is the stress exerted by the bed onto the ice (Pa), $c_t$ is the change in pressure melting point with temperature (K Pa$^{-1}$), and $c_w$ is the heat capacity of water (J kg$^{-1}$ K$^{-1}$). We note that the last term takes into account the changes in sensible heat due to pressure melting point variations. This term is often considered negligible and dropped from similar models (de Fleurian et al., 2018).





Finally, the effective drainage-system capacity $b'$ evolves according to opening and closure terms that are typically model-specific. As in Sommers et al. (2018), opening can be due to melt and sliding over bumps on the bed, while closing is solely due to ice creep:

$$\frac{\partial b'}{\partial t} = \frac{\dot{m}}{\rho_i} + \beta u_b - A|P_i - P_w|^{n-1}(P_i - P_w)l_c, \tag{6}$$

where $\rho_i$ is the ice density (kg m$^3$), $u_b$ is the magnitude of the sliding velocity (m s$^{-1}$), $A$ is the ice-flow parameter (Pa$^{-3}$ s$^{-1}$),
$n$ is the flow-law exponent (typically, $n = 3$) and $P_i$ is the ice overburden pressure (Pa). The parameter $\beta = max((b_r - b)/l_r, 0)$ is dimensionless; it governs opening by sliding and is a function of the typical bed bump height ($b_r$) and bump spacing ($l_r$) in such a way that opening by sliding only occurs where the gap height is less than the typical local bump height. The quantity $l_c$ is the creep length scale, which is defined as follows:

$$l_c = \begin{cases} b\left(1.0 - \frac{(b_c - b)}{b_c}\right), & \text{if } b \leq b_c, \\ b, & \text{otherwise,} \end{cases} \tag{7}$$

with $b_c$ a critical gap height controlling the creep (which is progressively cut off for $b \leq b_c$).

We do not allow for the drainage space to be partially filled, such that $b = b'$ always. Eqs. 1 and 6 can then be combined to produce an equation for the evolution of the hydraulic head:

$$\nabla \cdot \left[\frac{-b^3 g}{12\nu(1 + \omega Re)}\nabla h\right] + \frac{\partial b_e}{\partial t} = \dot{m}\left[\frac{1}{\rho_w} - \frac{1}{\rho_i}\right]$$
$$+ A|P_i - P_w|^{n-1}(P_i - P_w)l_c - \beta u_b + e_s. \tag{8}$$

## 2.2    Model parameters

Constants and parameters presented in the previous section are summarized in Table 1, along with typical values. Note that the englacial storage volume, $b_e$, present in the original set of equations (Sommers et al., 2018) is not currently used in SUHMO. Under this assumption, Eq. 8 becomes a standard elliptic partial differential equation (PDE).

### 2.3    Introduction of a diffusion term

As described in the introduction, the subglacial drainage system is made of various coexisting and dynamically evolving
subglacial drainage structures. These are typically broken into 2 categories: distributed (or inefficient) flow structures and channelized (or efficient) flow structures. The advantage of this set of governing equations is that only one equation (i.e., Eq. 6) is necessary to model the drainage space, and this equation accommodates both inefficient and efficient elements. One drawback from Eq. 6, however, is that it was originally derived by considering a sheet drainage system, central to which is the growth of the water sheet thickness, while equations governing channel growth are intrinsically two-dimensional. In practice,
adding a simple melt-opening term in Eq. 6 to accommodate channels causes the water sheet thickness to grow locally (e.g. in each computational cell), because the second dimension – the channel width, has been dropped from the problem formulation.





| Symbol | Description | Units | typical value |
|---:|---|:---:|:---:|
| $z_b$ | Bed elevation | m | N/A |
| $\boldsymbol{\tau}_b$ | Stress exerted by the bed onto the ice | Pa | 0.0 |
| $g$ | Gravitational acceleration | m s$^{-2}$ | 9.81 |
| $\rho_w$ | Bulk density of water | kg m$^{-3}$ | 1000 |
| $\rho_i$ | Bulk density of ice | kg m$^{-3}$ | 910 |
| $\nu$ | Kinematic viscosity of water | m$^2$ s$^{-1}$ | 1.787 x 10$^{-6}$ |
| $\omega$ | Transition between laminar and turbulent flow | - | 0.001 |
| $A$ | Ice flow-law parameter | Pa$^{-3}$ s$^{-1}$ | 2.5 x 10$^{-25}$ |
| $n$ | Ice flow-law exponent | - | 3 |
| $b_r$ | Typical height of bed bumps | m | 0.1 |
| $l_r$ | Typical spacing between bed bumps | m | 2.0 |
| $b_c$ | Creep cut-off length scale | m | 0.001 |
| $\mathbf{u}_b$ | Sliding velocity | m s$^{-1}$ | $(10^{-6}, 0)$ |
| $L$ | Latent heat of fusion of water | J kg$^{-1}$ | 3.34x10$^5$ |
| $G$ | Geothermal flux | W m$^{-2}$ | 0.0 |
| $c_t$ | Change of pressure melting point with temperature | kg Pa$^{-1}$ | 7.5 x 10$^{-8}$ |
| $c_w$ | Heat capacity of water | J kg$^{-1}$ K$^{-1}$ | 4.22 x 10$^3$ |
| $e_s$ | External meltwater source | m s$^{-1}$ | N/A |

**Table 1.** List of constants and parameters employed in SUHMO

This behavior is not an issue if relatively coarse meshes are used in configurations where the primary interest is effective pressure fields, but it does prevent channel geometries from converging with mesh resolution. Convergence with mesh resolution is an important feature of consistent numerical methods; additionally, AMR requires consistent convergence with mesh

resolution to be effective. As a first step towards addressing this issue, we modify Eq. 6 by adding a diffusion-like term, as follows:

$$\frac{\partial b}{\partial t} = \frac{\dot{m}}{\rho_i} + \beta u_b - A|P_i - P_w|^{n-1}(P_i - P_w)l_c + \nabla \cdot \mathcal{D}\nabla b, \qquad (9)$$

where the diffusion coefficient depends on the heat dissipation piece of the melt rate:

$$\mathcal{D} = \frac{b}{\rho_i L}(-\rho_w g \mathbf{q} \cdot \nabla h + c_t c_w \rho_w \mathbf{q} \cdot \nabla P_w). \qquad (10)$$

With this formulation, we aim to represent melting of channel walls as heat dissipation is no longer limited to channel/cavity ceilings. Adding this diffusion term and neglecting englacial storage, Eq. 8 then becomes:

$$\nabla \cdot \left[\frac{-b^3 g}{12\nu(1 + \omega Re)}\nabla h\right] = \dot{m}\left[\frac{1}{\rho_w} - \frac{1}{\rho_i}\right]$$

$$+ A|P_i - P_w|^{n-1}(P_i - P_w)l_c - \nabla \cdot \mathcal{D}\nabla b - \beta u_b + e_s. \qquad (11)$$





## 3    Algorithm details

We solve Eqs. 9 and 11 on a hierarchy of block-structured, Cartesian meshes using a finite volume discretization, facilitated by
the Chombo framework (Adams et al., 2001-2021). We extend the Chombo toolbox to solve the nonlinear evolution equation
for the hydraulic head implicitly using the Full Approximation Scheme (Briggs et al., 2000). The resulting algorithm is second-
order in space and first-order in time. For completeness, Appendix A gives a brief summary the main features of our AMR
framework. We adopt the notation used in several previous studies (Berger and Colella, 1989; Martin et al., 2008; Cornford
et al., 2013; Parkinson et al., 2020), and the reader is referred to these prior publications for additional details. All that follows
is specific to a two-dimensional application with an isotropic Cartesian mesh.

### 3.1    Full Approximation Scheme for variable coefficient, nonlinear elliptic PDE

#### 3.1.1    Multigrid methods

Multigrid (MG) methods are commonly employed to solve problems of the type

$$\mathcal{A}(u) = \mathcal{F}. \tag{12}$$

If we denote by $\tilde{u}$ an approximation to the exact solution of this problem, we can define the error $e$ as $e = u - \tilde{u}$ and the residual
$r$ as $r = \mathcal{F} - \mathcal{A}(\tilde{u})$. If $\mathcal{A}$ is a linear operator then we obtain the residual equation:

$$\mathcal{A}(e) = r, \tag{13}$$

with which we can start a recursive process of "restricting" (averaging) the residual onto a coarser grid to solve for a coarse
correction of the error that we then interpolate back (usually referred to as "prolongation") to the fine grid (Briggs et al., 2000).
For linear systems, multigrid is highly efficient, and can be implemented using a matrix-free approach. It is also straightforward
to extend to AMR mesh hierarchies (Martin and Cartwright, 1996).

#### 3.1.2    FAS Multigrid

If $\mathcal{A}$ is a nonlinear operator, we cannot make use of Eq. 13. Instead we use the residual in a new problem formulation:

$$\mathcal{A}(h) = r + \mathcal{A}(\tilde{h}), \tag{14}$$

with which we can follow a similar recursive process as for a traditional MG method. For completeness, the main steps of one
FAS MG iteration are described here and illustrated in Fig. 1, focusing on the V-cycle scheduling of events; a more thorough
description can be found elsewhere (Briggs et al., 2000; Henson, 2003). We have chosen to work with $h$ instead of $u$ to be
consistent with our main variable – the hydraulic head. We will use the following general expression for the nonlinear operator
$\mathcal{A}$:

$$\mathcal{A}(h) = (\alpha \mathbf{A} I - \beta \nabla \cdot \mathbf{B} \nabla) h + \mathcal{G}(h), \tag{15}$$

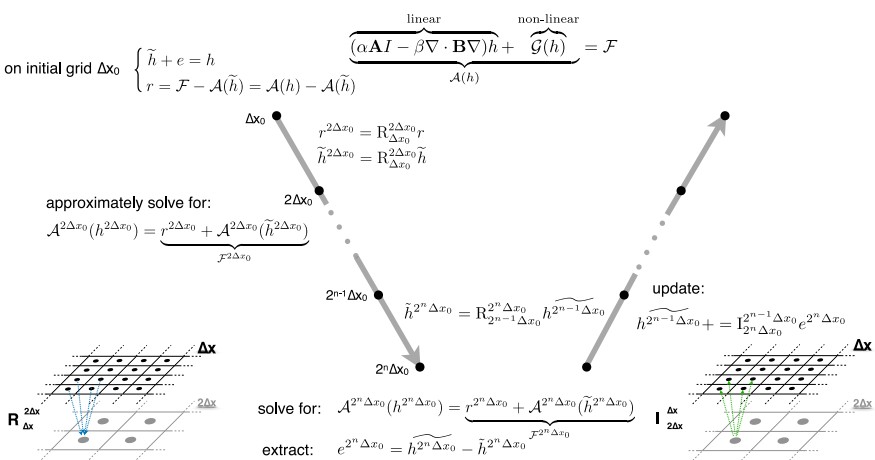

**Figure 1.** Illustration of one FAS MG iteration, following a V-cycle schedule of grids. See text for further explanations.

where the nonlinear piece is contained in $\mathcal{G}(h)$. The restriction process is illustrated on the left of Fig. 1, by the arrow pointing down, while the prolongation is illustrated on the right with the arrow going up. The original level is labelled the finest level, and it has a mesh spacing of $\Delta x_0$. This level is coarsened $n$ times by a factor of 2, so that coarsened grid $n$ has a grid size of $2^n \Delta x_0$.

We start by finding an approximate solution $\tilde{h}$ to the original problem Eq. 12 on the finest grid, by using a few iterations of an iterative method (the relaxation method is described in Appendix B). The residual $r$ is then evaluated, and both $\tilde{h}$ and $r$ are averaged down to the first coarsened grid, via a restriction operator denoted as $R^{2\Delta x_0}_{\Delta x_0}$. We rewrite the problem as in Eq. 14 (where the new right-hand side –RHS, is denoted $\mathcal{F}^{2\Delta x_0}$ in Fig. 1) and perform a number of relaxation iterations to find an approximate solution, $\widetilde{h^{2\Delta x_0}}$. This process is repeated until we reach the coarsest level. The coarsest error, $e^{2^n\Delta x_0}$,

can then be extracted from the solution and interpolated up to the next finer grid to be used to correct the local approximation $\widetilde{h^{2^{n-1}\Delta x_0}}$ (see Fig. 1). This "prolongation" process is repeated going up the levels until we reach the finest level again. We usually perform between 2 and 4 relaxation iterations while correcting the solution at each level.





### 3.1.3 Special treatment for the variable coefficient

We rewrite Eq. 11 to make it consistent with the operator that has been described in Fig. 1:

$$
\overbrace{\nabla \cdot \Big[ \overbrace{\frac{-b^3 g}{12\nu(1+\omega Re)}}^{\mathbf{B}} \nabla h \Big]}^{linear}
$$
$$
\overbrace{- A|P_i - \rho_w g(h-z_b)|^{n-1}(P_i - \rho_w g(h-z_b))l_c}^{nonlinear}
$$
$$
= \overbrace{-\frac{\rho_c}{L}\rho_w g \Big[ 1 - c_t c_w \rho_w \Big] \mathbf{q} \cdot \nabla h}^{nonlinear\ lagged}
$$
$$
+\frac{\rho_c}{L}(G + |\mathbf{u}_b \cdot \tau_b| - c_t c_w \rho_w^2 g \mathbf{q} \cdot \nabla z_b) - \nabla \cdot \mathcal{D}\nabla b - \beta u_b + e_s.
$$

(16)

We discuss our treatment of the remaining nonlinearity and $h$ dependencies in the RHS in Section 3.2. Note that the coefficient $\mathbf{B}$ is both spatially variable and a function of the primary variable $h$, via the coupling with $Re$. In our final implementation of the algorithm, $\mathbf{B}$ is recomputed on the finest grid at the beginning of every V-cycle, and then averaged down on all coarser grids. We experimented with fully lagging $\mathbf{B}$, estimating the Reynolds number before the first V-cycle iteration and freezing its value for the remaining of the solve, but this resulted in poor overall algorithmic efficiency, as discussed in Appendix C.

### 3.2 Map of the algorithm

In order to solve the coupled set of Eqs. 9 and 16, we use a combination of FAS-MG iterations as described in Section 3.1 to solve for the hydraulic head and traditional MG to solve for the gap height. We note that the lagged term in Eq. 16, as well as any term involving the water flux, will depend on $h$ in a non-trivial way. To ensure these are treated properly, we embed the FAS-MG solve for $h$ in external Picard iterations (in which the value of the gap height is frozen). We have found the required number of Picard iterations is typically 1 or 2.

We use a backward Euler method to discretize the temporal term in Eq. 9, reorganizing to be consistent with Eq. 12:

$$
\overbrace{(I - \Delta t \nabla \cdot \mathcal{D}\nabla)b^{n+1}}^{\mathcal{A}(b^{n+1})} = b^n + \Delta t\Big[ \frac{\dot{m}}{\rho_i} + \beta u_b - AN^3 l_c \Big],
$$

(17)

where we have replaced the effective pressure $(P_i - P_w)$ with $N$, and $n = 3$ has been assumed.

The main steps required to advance our main variables from $t = t^n$ to $t = t^{n+1}$ are summarized in Algorithm 1. The superscripts $n$ and $k$ refer to the current time-step and the current Picard iteration, respectively. For better readability, $n$ is omitted when discussing Picard iterations. When relevant, "CC" will refer to cell-centered variables, while "FC" will refer to face-centered variables.

Following common usage (for example, in Martin and Colella (2000)) we enforce boundary conditions at domain boundaries and between AMR grid patches (both at the same refinement level and between levels) through the use of a ring of ghost cells ("GC") around each logically rectangular patch.




---

**Algorithm 1** Skeleton of a SUHMO time step ($t^n \rightarrow t^{n+1}$)

---

**(I) Start the time step**

(a) Fill GC of $h^n$ and $b^n$

(b) $h_{old} \leftarrow h^n$ and $b_{old} \leftarrow b^n$

**(II) Evaluate $h^{n+1}$**

**while** ! converged **do**

(a) $h_{lagged} \leftarrow h_k$

(b) Compute $\nabla h_k^{cc/fc}$ and $\nabla z_{b_k}^{cc/fc}$

(c) $Re$-**q** dependency

   - Compute $Re_k^{cc}$ by solving the quadratic equation $\omega Re^2 + Re - \frac{b^3 g}{12\nu^2}(|\nabla h|) = 0$

   - Evaluate $Re_k^{fc}$

   - Update $\mathbf{q}_k^{fc}$ based on Eq. 2

   - Evaluate $\mathbf{q}_k^{cc}$

(d) Compute the external source term $e_{sk}$ based on the type of external water input (localized/distributed)

(e) Compute the RHS of Eq. 16

   - Update $\dot{m}_k$ using Eq. 5. All dot products are computed at FC before being interpolated to CC

   - Evaluate the FC diffusive coefficient based on Eq. 10, $\mathcal{D}_k = \mathrm{f}(\nabla h_k^{fc}, \nabla z_{b_k}^{fc}, \mathbf{q}_k^{fc})$

(f) Solve for $h_{k+1}$ (Eq. 16) using the FAS-MG method described in Sect. 3.1

(g) Average down and fill GC of $h_{k+1}$

(h) Check for convergence

**if** $\frac{\|h_{k+1} - h_{lagged}\|_\infty}{\|h_{old}\|_\infty} \leq \epsilon$ **then**

   $h^{n+1} \leftarrow h_{k+1}$

   $converged = true$

**end if**

**end while**

**(III) Evaluate $b^{n+1}$**

(a) Re-evaluate $\nabla h^{n+1,cc}$, $Re^{n+1,cc/fc}$, $\mathbf{q}^{n+1,cc/fc}$ based on $h^{n+1}$

(b) Compute the RHS of Eq. 17

   - Update $\dot{m}^{n+1}$ based on Eq. 5

(c) Solve for $b^{n+1}$ (Eq. 5) using a traditional MG method with a Gauss-Seidel relaxation method

   - Re-evaluate $\mathcal{D}^{n+1}$ using Eq. 10 and hold fixed during the solve

**(IV) End the time step**

(a) Write a plot/checkpoint file or perform post processing analysis

---





## 4    Analysis of the algorithm efficiency

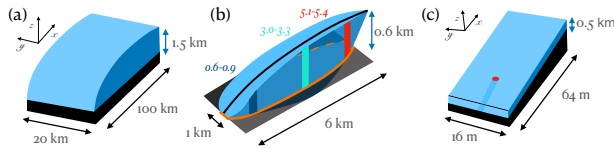

**Figure 2.** (a) and (b) are sketches of synthetic glacier topographies used in SHMIP (de Fleurian et al., 2018): (a) 100 km long ice sheet margin with a maximum thickness of 1500 m, and (b) 6 km long valley glacier with a 600 m altitude difference between summit and terminus. The orange region in (b) outlines the intersection of $z_b = 0$ with the ice – note the bed overdeepening. (c) is a sketch of the channelizing test case geometry used for convergence analysis. Note the thickness of the ice. The red dot shows the location of the moulin, which has a diameter of 1.5 m.

### 4.1    Distributed test case

We start by analyzing a simple distributed-flow test case. The topography considered is the synthetic representation of a land-terminating ice sheet margin from SHMIP (de Fleurian et al., 2018), illustrated in Fig. 2 a). The ice-sheet domain measures 100 km in the $x$ direction and 20 km in the $y$ direction, the bed is flat and a parabolic ice surface $z_s$ is prescribed by:

$$z_s(x,y) = 6(\sqrt{x + 5000} - \sqrt{5000}) + 1. \tag{18}$$

In order to evaluate the convergence properties of the complete algorithm, we evolve the system to a fixed time with increasing resolution, halving $\Delta x$ with each refinement. The solution error is then computed by comparing $\Phi^c$, the solution at resolution $\Delta x$, with the finer solution $\phi^f$ computed using $\Delta x/2$ and averaged onto the coarser grid ($\Phi^{f \to c}$). The $L^2$ norm of the error for a simulation with $n_{tot}$ cells is:

$$Err_{L^2} = \sqrt{\frac{1}{n_{tot}} \sum_{i=0}^{n_{tot}} (\Phi_i^c - \Phi_i^{f \to c})^2}. \tag{19}$$

Figure 3 a) shows the error convergence using six grid resolutions, for 2 variables of interest: $h$ and $P_w$. Note that since $P_i$ is constant, the convergence of the effective pressure $N$ will be similar to that of $P_w$. The slope of the error demonstrates second-order convergence for both variables.

### 4.2    Channelized test case

We now turn our attention to a channelizing test case. The domain is a rectangle of 64 m in the $x$ direction by 16 m in the $y$ direction. The bed is sloped in the $x$ direction (with a +2% slope) and is topped with a slab of ice of constant thickness everywhere (500 m). A moulin delivering 30 m$^3$ s$^{-1}$ of water is located 16 m from the outlet of the domain in the $x$ direction. The geometry is shown in Fig. 2 c). Periodicity is assumed in the $y$ direction, a homogeneous Neumann boundary condition

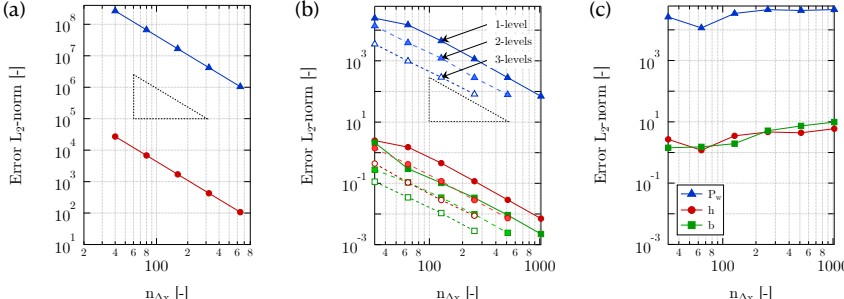

**Figure 3.** Convergence results for (a) the distributed test case, (b) the channelized test case with the diffusion term and (c) the *original* channelized test case without diffusion. The variables shown are the head ($h$) in red, the water pressure ($P_w$) in blue and the gap height ($b$) in green. The $x-$axis is the number of cells in the $x$ direction, $n_{\Delta x}$

.

(outlet) is prescribed at x=0 m while a homogeneous Dirichlet boundary condition is prescribed at x=64 m. Note that the moulin source term follows a Gaussian distribution (see the top plot of Fig. 4 a)), the convergence of which is exactly second order.

We use Eq. 19 to evaluate the convergence properties of SUHMO on this slightly more challenging test case. Results using
seven grid resolutions are presented in Fig. 3 b-c). The *original* formulation (Eqs. 6 and 8), fails to converge: as the resolution increases, the channel width becomes smaller and smaller, with the limit being the cell size $\Delta x$. This phenomenon is illustrated in the middle plot of Fig. 4 a) and in Fig. 4 b), for a simulation with $\Delta x = 0.125$ m. In this case, the gap height is seen to exceed 12 m, a situation deemed unphysical in most cases, and we clearly observe that all the flow is routed through a single cell. As can be seen in Fig. 3 b), adding the diffusion term to the formulation (Eqs. 9 and 16) enables second order convergence of all
the variables of interest. In this case, as is evidenced by the bottom plot of Fig. 4 a) and by Fig. 4 b), the channel width can extend over several $\Delta x$ and the overall shape and aspect ratio of the conduit better fits the idea of what a channel should be.

To demonstrate our AMR implementation, we also perform a numerical convergence analysis using several levels of refinement. The AMR scheme should ideally produce solutions with comparable accuracy to a uniform mesh solution with the same (finest) resolution. Using the previously described test case, a refinement criteria based on the local melting rate enables
refinement of the region where channelization occurs. Starting from a baseline, single-level simulation with a cell size of $\Delta x$, we enable the computation to continue and allow either one or two extra level(s) of refinement, where the finest level will have a cell size of either $\Delta x/2$ or $\Delta x/4$, respectively. This two- or three-level simulation is then compared to results at a finer resolution $\Delta x/4$ or $\Delta x/8$, respectively. The results shown in Fig. 3 b) indicate that the error using two or three AMR levels is comparable to that of the single-level solution with the same effective resolution. The entire plot can be read horizontally,
where an imaginary line drawn from a point on the *1 level* line should intersect the corresponding equivalent-resolution on the *2 levels* or *3 levels* line, as is the case here.



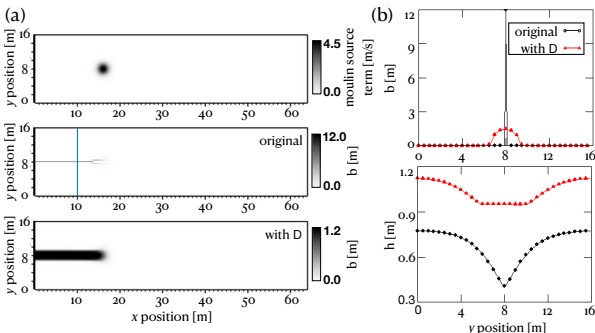

**Figure 4.** a) Two-dimensional fields of moulin input and gap height ($b$) extracted from computations with mesh size $\Delta x$ = 0.125 m. b) One dimensional plots of gap height ($b$) and head ($h$) extracted from the two dimensional fields, at location $x$ = 10 m, highlighted by the blue line in a). Note that *original* refers to the formulation without the diffusion term, while *with D* refers to the formulation including the diffusion term.

## 5 SHMIP suite of test cases

Having demonstrated the convergence and efficiency of SUHMO, we turn our attention to a set of simple test cases from the Subglacial Hydrology Model Intercomparison Project (SHMIP) (de Fleurian et al., 2018) to demonstrate the validity of our
implementation. SHMIP is built around six synthetic *Suites* of experiments (labelled from A to F), each consisting of a set of four to six numerical experiments, designed to show the formation and evolution of the different drainage elements (sheets and channels) in the context of different input scenarios. Two geometries are considered, a land-terminating ice sheet margin and a synthetic valley-glacier geometry, as shown in Fig. 2 a) and b), respectively. In the paper, the results from 13 different models are presented, including the SHAKTI model from Sommers et al. (2018).

We show steady-state cases (Suites A, B and E) in Sect. 5.1, and use Suite F in Sect. 5.2 to explore seasonal forcing. We compare all of our results to those obtained with SHAKTI (all results from the SHMIP project are open source and freely available online, see Werder et al. (2018)). Note that the diffusion term is included in what follows, but its impact is negligible in these non-channelizing test cases. Additionally, the creep cut-off length scale $b_c$ is set to 0 so that the creep length scale $l_c$ reverts to $b$ (see Eq. 7), consistent with the SHAKTI contribution in de Fleurian et al. (2018).

### 5.1 Steady-state test cases

Results for the SHMIP test cases A, B and E are presented in Figs 5, 6 and 7, respectively. The longitudinal evolution of the width-averaged effective pressure $N$ is displayed on the left hand side of all figures. The right hand side shows the total *discharge* and its various contributions, to be compared with the total *recharge*. The discharge is the evolution along the $x$-axis of the $y$-integral of the face-centered water flux $\mathbf{q}^x$; while the volumetric recharge contains contributions from both moulin
input (when applicable) and melting. Using the notation introduced in Sect. A2, the discharge at each $x_i$ location can be





computed on the coarsest grid as:

$$dis(x_i) = \sum_{\substack{\mathbf{p} \in (i, \mathbb{Z}) \\ \mathbf{p} \in \Omega^{0,x}}} \mathbf{q}_\mathbf{p}^x \Delta x^0, \tag{20}$$

while the recharge is the cell-integrated right hand side of Eq. 1):

$$rech(x_i) = \sum_{\substack{\mathbf{p} \in (i, \mathbb{Z}) \\ \mathbf{p} \in \Omega^0}} \left( \frac{\dot{m}_\mathbf{p}}{\rho_w} + e_{s,\mathbf{p}} \right) \Delta x^0 \Delta x^0. \tag{21}$$

At steady state, the recharge and discharge at the domain outlet should be exactly the same. Some figures also display *efficient* and *inefficient* contributions to the total discharge, which are computed based on a Degree of Channelization (DoC) variable. The DoC is a cell-centered variable used to quantify the relative contribution of the two ice-opening terms in the the RHS of Eq. 9:

$$DoC = \frac{\frac{\dot{m}}{\rho_i}}{\frac{\dot{m}}{\rho_i} + \beta u_b}, \tag{22}$$

with values between zero and one in each cell. A DoC close to one indicates a high degree of channelization, while a value close to zero is indicative of a sheet-like drainage system. The efficient and inefficient contributions to the total discharge at $x_i$ are expressed as $dis(x_i)DoC(x_i)$ and $dis(x_i)(1 - DoC(x_i))$, respectively.

Suites A and B use the land-terminating ice sheet margin geometry (Fig. 2 a)). In Suite A, a steady and spatially uniform water input is prescribed, with total recharge increasing as we progress from A1 to A6. In Suite B, the same amount of water

as in case A5 is fed into an increasing number of moulins (from one in B1 to a hundred in B5). For additional details on the parameterization of the different Suites, the reader is referred to the online instructions [1]. Single-level (no AMR) simulations are performed, with a fixed cell size $\Delta x^0$ = 312.5m and a fixed time step $dt$ = 1h, which is consistent with values reported from other two-dimensional models in de Fleurian et al. (2018). We apply a Dirichlet boundary condition $h = z_b$ at the left edge (outlet) of the domain, and Neumann boundary conditions with 0 prescribed flux on all other domain boundaries. The

steady state is quickly reached in all cases, and simulations are run for approximately 400 days. Results obtained with SUHMO compare well to those obtained with SHAKTI, as expected since both models are built on the same set of equations. We note from Fig. 5 b) and Fig. 6 b) that SUHMO exhibits slightly smaller contributions from the effective system on cases that display a hybrid flow configuration (see A6 in Fig. 5 b) for example). These discrepancies are attributed to discretization differences between both models. SHAKTI uses an unstructured mesh while SUHMO uses regular Cartesian meshes. We

note that discretization features are probably the cause of the overshoot observed in Fig. 6 b) for case B1 with SHAKTI. The recharge provided by each moulin in Suite B is another potential source of inconsistency. In this study, a Gaussian distribution is assumed.

Suite E is designed to investigate the effect of bedrock slope, and uses the synthetic valley glacier geometry (Fig. 2 b)). In this experiment, water input is uniformly distributed at the bed of the glacier. Here also, single-level simulations are performed,

---

[1]https://shmip.bitbucket.io/instructions.html





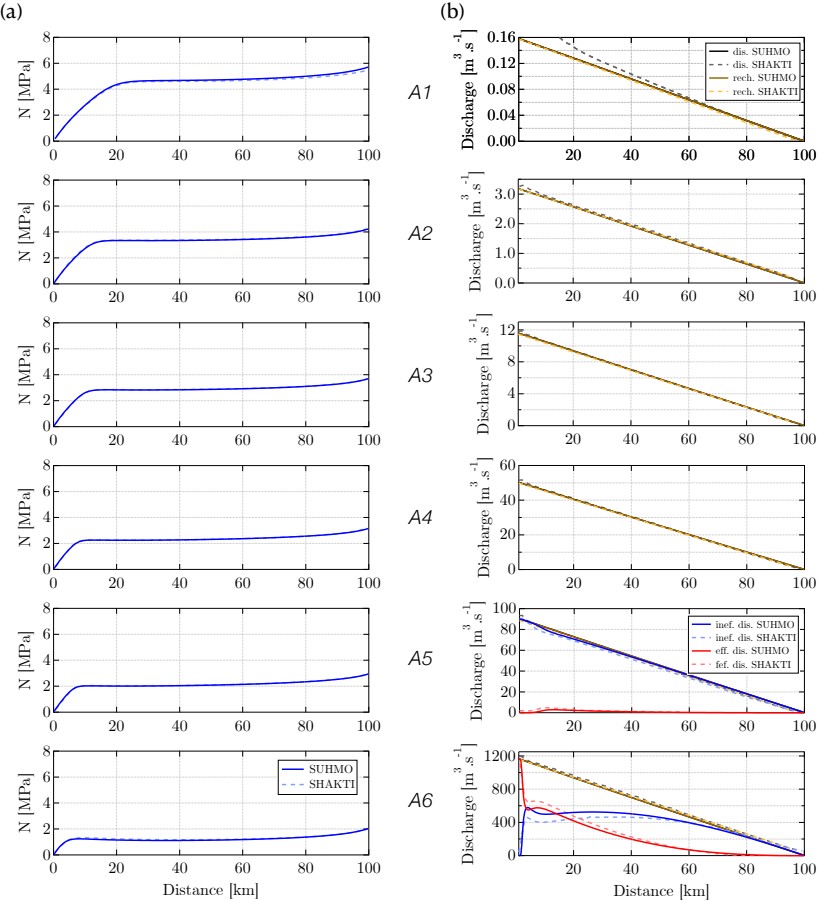

**Figure 5.** Steady-state results for the SHMIP suite of test cases A: (a) $y$-average evolution of effective pressure $N$ with distance from the outlet and (b) volumetric discharge, to be compared to the total recharge the system receives (see text for further explanation). Cases A5 and A6 also display the contributions from the inefficient and efficient systems (their sum gives the total discharge). For validation purposes, results from SHAKTI (Sommers et al., 2018) as presented in de Fleurian et al. (2018) are also shown.

.

with $\Delta x^0$ = 23.4375 m and $dt$ = 1h. The boundary conditions are similar to those in Suites A and B. Note that the value of $c_t$ is set to 0 for this Suite of experiments (removing the pressure-melt term in Eq. 5). The steady state is quickly reached in all cases, and simulations are run for approximately 400 days. While the effective pressure distributions obtained using SUHMO compare well to those obtained using SHAKTI, we note bigger differences in the spatial distribution of the hybrid flow configuration in Fig. 7 b). These are again attributed to differences in mesh and cell size between SUHMO and SHAKTI.

We note the same tendencies as the over-deepening of the valley increases (from E1 to E5), however, with a more and more sheet-like distribution throughout.



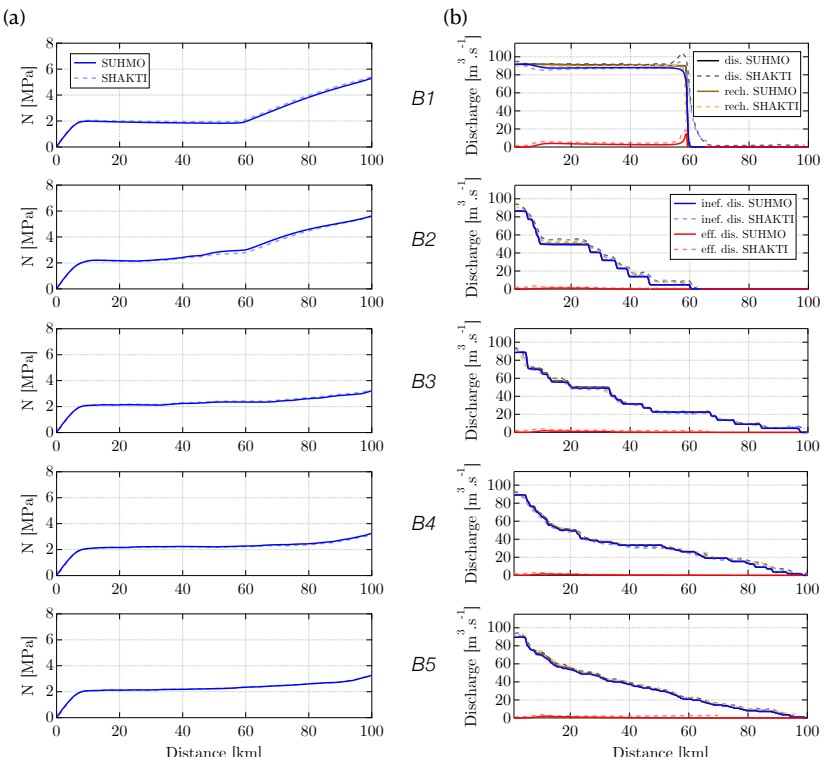

**Figure 6.** Steady-state results for the SHMIP suite of test cases B: (a) $y$-average evolution of effective pressure $N$ with distance from the outlet and (b) volumetric discharge, to be compared to the total recharge the system receives (see text for further explanation). The contributions from the inefficient and efficient systems (their sum gives the total discharge) are also featured. For comparison, results from SHAKTI (Sommers et al., 2018) as presented in de Fleurian et al. (2018) are also shown.

.

## 5.2 Suite F: seasonal cycle with valley topography

We turn our attention to SHMIP Suite F. The results presented in Sect. 5.1 focused on the effects of the geometry and water input type on otherwise steady-state configurations. Suite F, prescribes a seasonal water forcing in the synthetic valley glacier geometry of case E1 (Bench Glacier reference geometry). The water input increases from run F1 to run F5, as can be seen on Fig. 8 b). The setup follows that of Suite E. A total of six years are simulated, allowing sufficient time to settle into a periodic state. Year six results are presented in Fig. 8 a). Time evolutions of the averaged effective pressure $N$ are extracted at three locations of interest, labeled *low, middle,* and *high* bands, depicted in color in Fig. 2 b). As before, results obtained with SUHMO closely follow those of SHAKTI.

Overall, while comparisons with SHMIP do not enable a true validation of our results, they do help validate our algorithm and provide an idea of how SUHMO compares to other subglacial hydrology models available in the literature.



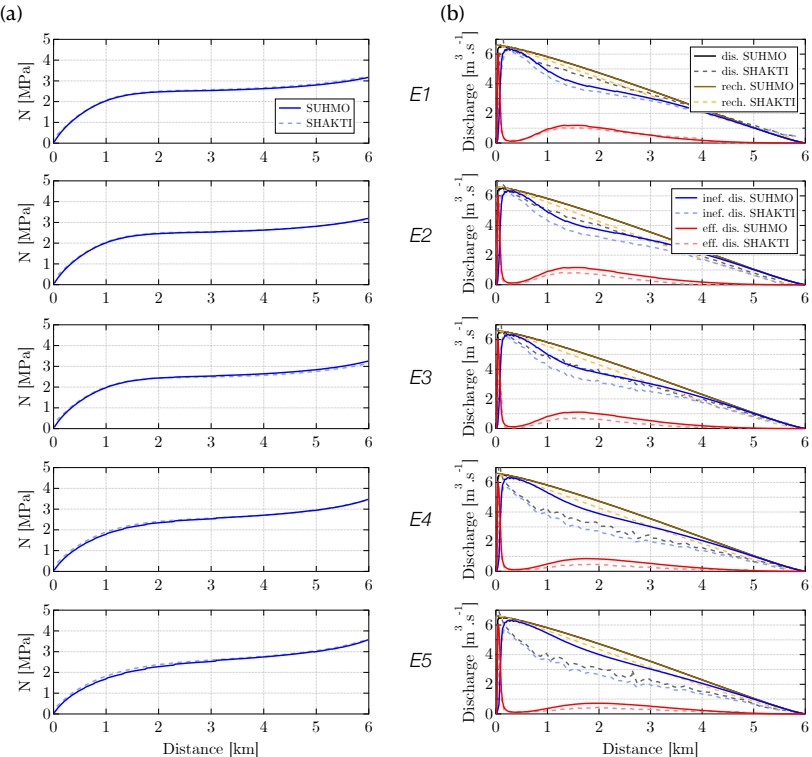

**Figure 7.** Steady-state results for the SHMIP suite of test cases E: (a) $y$-average evolution of effective pressure $N$ with distance from the outlet and (b) volumetric discharge, to be compared to the total recharge the system receives (see text for further explanation). The contributions from the inefficient and efficient system (their sum give the total discharge) are also featured. For validation purposes, results from SHAKTI (Sommers et al., 2018) as presented in de Fleurian et al. (2018) are also plotted.

.

# 6 AMR synthetic experiment

## 6.1 Case description

The test cases in Sect. 5 were all single-level experiments. In the present section, we now consider a synthetic square topography
of 100 km by 100 km, generated with the intent of emulating catchment areas found at ice sheet margins. Both the bed geometry and ice thickness are shown in Fig. 9 a). The bed height varies from 0 m to just under 1000 m, while the ice thickness increases from 100 m at the bottom left corner (red dot location) to 700 m at the top right corner. Zero flow via a homogeneous Neumann boundary condition is imposed on the two interior boundaries ($y = 0$ and $x = 100$ km) while Dirichlet boundary conditions are prescribed on the two other boundaries (with $h = z_b$, so that $P_w = 0$).

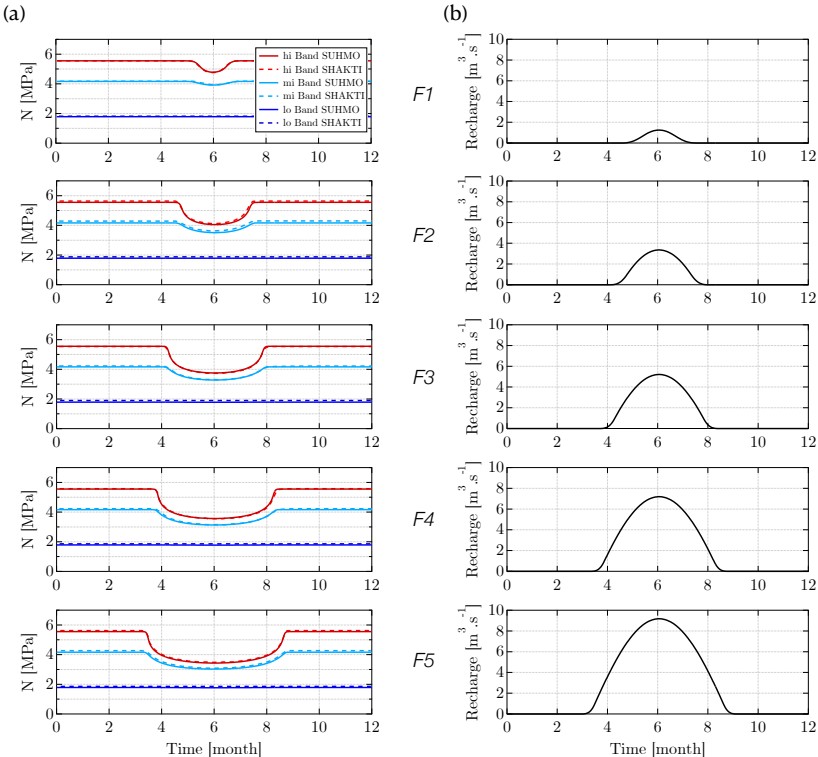

**Figure 8.** Steady-state results for the SHMIP suite of test cases F: (a) time evolution of the averaged effective pressure $N$ at three locations of interest (see text and Fig. 2 b)) and (b) time evolution of the seasonal water forcing. For comparison, results from SHAKTI (Sommers et al., 2018) as presented in de Fleurian et al. (2018) are also shown.

.

Five runs will be presented hereafter – labelled from $R_0$ to $R_4$ (see Table D1), all using a base mesh with 256 cells in both the $x$ and $y$ directions. This value is chosen so $\Delta x^0$ = 390.625 m, which is typical of many ice sheet simulations. The runs are forced by 63 randomly placed moulins, delivering a total water input of 5180 m³/s. The location of the moulins is shown in Fig. 9 b). We emphasize that the topography, moulins location and amount of water input used here have not been designed to reproduce an existing glacier area. The water delivered by the moulins is constant, no seasonal cycle is considered, and is

also quite high: this experiment should be taken as a demonstration of the robust behavior of the system even under prolonged high melting scenarios, when a high degree of channelization is expected. Our purpose is to demonstrate the importance of spatial resolution when looking at subglacial water patterns, and ultimately to examine the impact of resolution on the effective pressure distribution.

All runs start from an established, steady state, single-level simulation with no moulins. The moulins activate at $t = 0$ and

the influx ramps up over a period of 2 months until the maximum is reached. The simulations are run for another 22 months





after that, bringing the total simulated time to 2 years. The time step is fixed at 2 hours for the first year, before increasing to 5 hours for the remainder of the simulation.

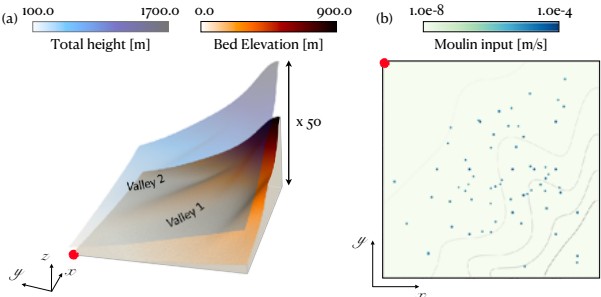

**Figure 9.** Synthetic square topography for runs $R_0$ to $R_4$ (a) extruded bed elevation (showing two valley regions) and ice height and (b) location of the moulins and isocontours of bed elevation. The red dot in both images shows the location of the lowest elevation – the outlet.

.

## 6.2 Overview of Computational requirements

An example of mesh configuration for runs $R_0$ to $R_3$ is shown in Fig. 10 b). In every case, a regridding operation is performed
each simulated week, so that the dynamic meshing can follow the water patterns and add or remove refinement around the channels as they develop or retract. We use both the gap heights and melting rates to tag cells for refinement. As can be seen, this criterion is very efficient, and only a small area of our entire computational domain ends up requiring up to 4 levels of refinement (due to the very small cell size reached, $R_4$ does not provide any additional visual insight and is therefore omitted from the figures).

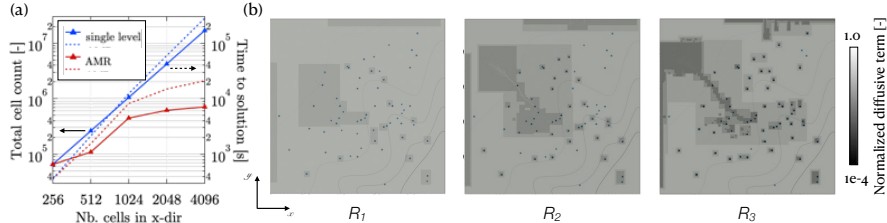

**Figure 10.** (a) Log-log plot of cell-counts (plain lines with triangles) and execution times (dotted lines) vs. resolution for the synthetic experiment and (b) Example of mesh distribution for runs $R_1$ to $R_3$ overlayed on the location of the moulins, normalized diffusive term (Sect. 2.3), and isocontours of bed elevation.

.

A single level run with the same resolution as $R_4$ (4096x4096) would require evolving over 16 million cells, when 20 times fewer cells are required in the AMR run to capture the important features, as can be seen in Fig. 10 a) (the cell count in the AMR




runs is obtained by averaging the total number of cells evolved over the course of the simulation). The total time to solution
is also shown in Fig. 10 a). For completeness, results from single-level simulations with resolution matching each finest level
of refinement $n$ of $R_n$ are also reported. The total time to solution in this case does not scale exactly with the increase in
resolution: the simulation contains a transient ramp up where moulins activate, and refining means creating more channels and
spatial stiffness, as evidenced by the increasing number of FAS MG iterations (not reported here). The time to solution ratio
of single level versus AMR simulations, however, increases with refinement and is well over an order of magnitude for the
most refined case presented here. Additionally, note that despite being performed very frequently, the cost of regridding never
accounts for more than 0.7% of the total computational time.

## 6.3 Results and analysis of the effective pressure distribution

The first three rows of Fig 11 show fields of gap height and effective pressure $N$ for runs $R_0$ and $R_2$ to $R_4$. A close up on the
valley 1 area (see Fig. 9) illustrates the extent and shape of the central channel. One interesting feature is that no channel forms
in $R_0$; no channelization ever occurs, even if the simulation is extended for another 10 years. $R_1$ is very similar to $R_0$ in that
no real channel inception can be seen, and is therefore omitted in Fig. 11. This appears to demonstrate a minimum resolution
to sufficiently resolve channelization behavior in this example. Similar minimum resolution requirements were demonstrated
for marine ice sheets in Cornford et al. (2016).

The bottom row of Fig 11 displays $N$ differences between $R_0$ and, respectively, $R_2$, $R_3$ and $R_4$, from left to right. As
expected, the effective pressure increases significantly in the valley 1 area and near the top boundary ($y = 100$ km) for $R_{n,n>1}$,
where channels are seen to develop. For $R_3$ and $R_4$, effective pressure differences can reach up to 10% of the global maximum
of $N$ and, more importantly, up to 25% of the local $N$ value (up to 0.6 MPa). These differences are deemed non-negligible
in the context of evaluating a locally varying friction law, such as the ones from Schoof (2005) or Tsai et al. (2015). Figure 1
in Brondex et al. (2017), for example, shows strong nonlinearities in certain low-pressure regimes, where a difference of this
order could result in very different basal drag evaluation. Such sensitive areas are more likely to be located near the grounding
line, where ice is thinner, and the potential impact on ice velocity warrants further investigation.

## 7 Concluding remarks

In this paper, we present and validate a novel AMR subglacial hydrology model, SUHMO, based on the Chombo frame-
work (Adams et al., 2001-2021). We solve equations similar to those in Sommers et al. (2018), with the addition of a pseudo-
diffusion to recover the wall melting in channels that was discarded in the derivation of the original equations. We demonstrate
the usefulness of this additional term in achieving consistent spatial convergence as finer resolution begins to resolve flow con-
figurations. Our algorithm uses an efficient combination of nonlinear MG iterations, embedded in external Picard iterations.
We show that results with SUHMO closely follow those obtained with SHAKTI (Sommers et al., 2018) on a broad selection
of the SHMIP suite of test cases (de Fleurian et al., 2018). A more complex, multi-level test case is also presented; compu-
tational performance analysis demonstrates the efficiency of AMR on such large-scale hydrologic problems, when compared

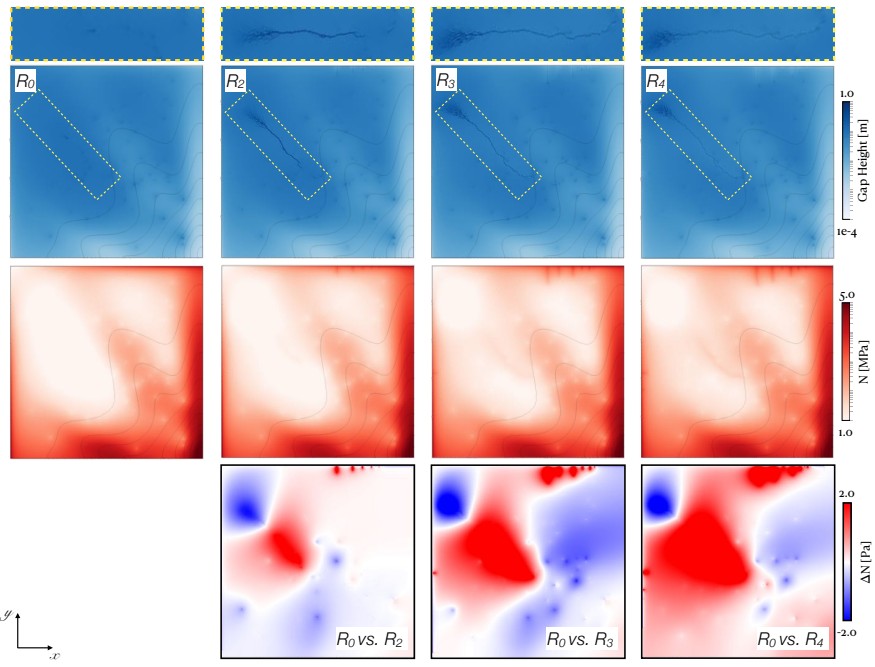

**Figure 11.** Fields of gap height (top two rows) and effective pressure (third row) for runs $R_0$ and $R_2$ to $R_4$, with overlayed isocontours of bed elevation. For each run, a close-up of the channelizing area in the valley 1 region (see Fig 9) is also displayed. The bottom row shows fields of effective pressure differences between $R_0$ and $R_2$ to $R_4$, from left to right.

.

to a single-level run with the same spatial discretization. The AMR approach will eventually enable better ice-bed boundary conditions for ice sheet simulations at a reasonable computational cost.

With that in mind, future work will focus on the coupling of SUHMO with the BISICLES AMR ice sheet model (Cornford et al., 2013), in order to further investigate the sensitivity of model predictions to basal conditions. Indeed, while the precise topography of the subglacial network is generally deemed unimportant to the overall ice sheet dynamics, there is to our knowledge no real numerical proof of this assessment. A numerical tool capable of resolving the structure of channels, following them as they emerge and disappear would be an asset in helping to determine if this is indeed the case.





*Code availability.* We used the GMD_release branch of the publicly available version of the SUHMO subglacial hydrology model: `https://github.com/EnnaDelfen/SUHMO/branches` SUHMO is written in a combination of C++ and FORTRAN and is built upon the Chombo AMR software framework. More information about Chombo may be found at `http://Chombo.lbl.gov`.





## Appendix A:  AMR structure and notation

### A1    Proper nesting

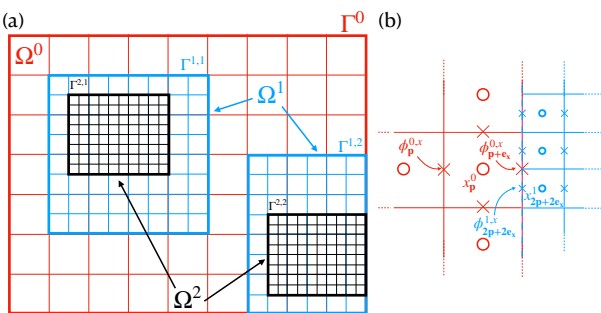

**Figure A1.** a) Example of a block structured mesh composed of 3 levels. Discrete level domain $\Omega^0$ comprises the cell centers of the coarsest grid, $\Gamma^0$. Level domains $\Omega^1$ and $\Omega^2$ are each built from two separate rectangular blocks, each with their own separate grids. b) Focus on coarse-fine interface between $\Omega^0$ and $\Omega^1$. Location of cell- and face- centered data are represented with circles and crosses, respectively. Face-centered data belonging to $\Omega^{0,x}$ on the interface are replaced by averaging of $\Omega^{1,x}$ (finer) data.

Calculations are performed on a hierarchy of $\ell_{max}$ nested, cell-centered level domains. For each AMR level $\ell = 0, ..., \ell_{max}$, the problem domain $\Omega^\ell$ is discretized by a uniform Cartesian grid $\Gamma^\ell$ with grid spacing $\Delta x^\ell$. Level 0 is the coarsest level, encompassing the entire geometry, while each subsequent finer level, $\ell + 1$, is a factor $n^\ell_{ref} = \frac{\Delta x^\ell}{\Delta x^{\ell+1}}$ finer than level $\ell$ ($n^\ell_{ref}$ is a power of 2, usually 2). Each $\Omega^\ell$ is constructed from one or more rectangular subsets of $\Gamma^\ell$, as can be seen in Fig. A1 a): $\Omega^1$, for example, is built from two separate rectangular blocks, each with their own subgrid $\Gamma^{\ell,*}$. An important property is that each domain level is properly nested; that is, no interfaces exist between $\Omega^\ell$ and $\Omega^{\ell\pm2}$; only between two subsequent levels or the domain boundary.

Certain derived quantities, such as fluxes, are located on two supplementary hierarchies of face-centered level domains, that will be denoted $\Omega^{\ell,x}$ and d $\Omega^{\ell,y}$ for x- and y- centered faces, respectively, on level $\ell$.

### A2    Cell and face centered data

Variables can be cell-centered or face-centered. We define a grid vector, $\mathbf{p} \in \mathbb{Z}^2$, choosing to number cells starting at (0,0), and grid basis vectors $\mathbf{e}_x = (1,0)$ and $\mathbf{e}_y = (0,1)$. Cell centers within $\Omega^\ell$ are then located at $\mathbf{x}^\ell_{\mathbf{p}} = \Delta x^\ell(\mathbf{p} + \frac{1}{2}(\mathbf{e}_x + \mathbf{e}_y))$ and the midpoints of cell faces within $\Omega^{\ell,*}$ at $\mathbf{x}^\ell_{\mathbf{p}} \pm \frac{\Delta x^\ell}{2}\mathbf{e}_*$. Cell-centered level variables $\phi^\ell_{\mathbf{p}} = \phi(\mathbf{x}^\ell_{\mathbf{p}})$ and face-centered level variables $\phi^{\ell,*}_{\mathbf{p}} = \phi(\mathbf{x}^\ell_{\mathbf{p}} - \frac{\Delta x^\ell}{2}\mathbf{e}_*)$ follow naturally. Notice that $\phi^{\ell,x}_{\mathbf{p}}$ is located on the 'west' face of the cell $\mathbf{p}$ and $\phi^{\ell,y}_{\mathbf{p}}$ is located on the 'south' face of the cell $\mathbf{p}$.





### A3    Coarsening operator

We identify cells at different levels which occupy the same geometric regions by means of the coarsening operator $\mathcal{C}_r(\mathbf{p}) = \frac{\mathbf{p}}{r}$ and its inverse, the refinement operator. In that sense, $\mathcal{C}_r^{-1}(\mathbf{p})$ is the set of all cells in a grid $r$ times finer that represent the same geometric region (in a finite volume sense) as the cell $\mathbf{p}$ (Martin and Colella, 2000).

### A4    Composite variables

Discrete representations of continuous fields are then cell- and face- centered composite variables $\phi^{comp}$, made up from "valid" (or uncovered by a finer level) portions of the level variables. First, level domains $\Omega^{\ell}$ are divided into valid ($\Omega^{\ell}_{valid}$) and invalid ($\Omega^{\ell}_{invalid}$) regions, such that $\Omega^{\ell}_{valid} = \Omega^{\ell} - \mathcal{C}_{n^{\ell}_{ref}}(\Omega^{\ell+1})$. Valid level domains for face-centered quantities are defined in the same way, $\Omega^{\ell,*}_{valid} = \Omega^{\ell,*} - \mathcal{C}_{n^{\ell}_{ref}}(\Omega^{\ell+1,*})$. A composite variable is then defined on the union of all valid regions, $\boldsymbol{\Omega} = \bigcup_{\ell} \Omega^{\ell}_{valid}$, where $\phi^{comp}(\boldsymbol{\Omega}) = \bigcup_{\ell} \phi^{comp}(\Omega^{\ell}_{valid})$. Likewise, composite vector fields are valid on all faces not overlain by finer faces.

We also construct ghost regions $\Omega^{\ell}_{ghost}$ that surround $\Omega^{\ell}$. These usually contain one or two extra cells and exist purely for numerical convenience – to compute gradients or other face-centered quantities. These buffer regions contain either boundary specified values, or are used to store extrapolated data or data calculated by interpolation from valid regions of a coarser level. Details pertaining to the computation of composite operators such as gradients and Laplacians, can be found in prior publications (Martin and Colella, 2000).

### A5    Level variables – averaging down

It is sometimes necessary to transfer information from finer grids to coarser ones: $\mathcal{C}_{n^{\ell}_{ref}}(\Omega^{\ell+1})$ is typically filled from appropriate cell-centered (or face-centered) arithmetic averaging of level $\ell+1$ data. An example case is illustrated in Fig. A1 b): where level 0 and 1 meet, face centered quantity $\phi^{0,x}_{\mathbf{p}+1}$ would be replaced by $\frac{\phi^{1,x}_{\mathbf{2p}+2\mathbf{e_x}} + \phi^{1,x}_{\mathbf{2p}+2\mathbf{e_x}+\mathbf{e_y}}}{2}$.

### A6    Regridding

We regrid every $n_{regrid}$ timesteps. $n_{regrid}$ is typically fixed at the beginning of a run. During this process, the solution at each grid cell and on each level, whether valid or currently covered, is tested against some specified criteria (or a combination of) to determine if refinement is required, in which case the cell is tagged for refinement. A new set of grids is then generated to ensure all tagged cells are covered by a finer level, whilst still satisfying the rules introduced above regarding proper nesting. This procedure enables the refinement or coarsening of the grids over time, following regions of interest. The appropriate refinement criteria varies depending on the type of application. In the case of SUHMO, we typically refine based on high values of the melting rate, and/or gap height to ensure we resolve the channelization process.





## Appendix B:  Relaxation method in the FAS-MG algorithm

To relax Eq. 14 on each FAS level, we employ a nonlinear Gauss-Seidel with multicolor ordering. With the formulation of
Eq. 15 for the nonlinear operator, and using the notation introduced in Section A, we obtain the following discretized equation
for each $\mathbf{p}$ cell on a given level -the notation $\ell$ has been omitted:

$$
\alpha \mathbf{A_p} h_{\mathbf{p}}
$$
$$
+ \frac{\beta}{\Delta x^2} \Big[ \mathbf{B}^x_{\mathbf{p}+\mathbf{e}_x}(h_{\mathbf{p}+\mathbf{e}_x} - h_{\mathbf{p}}) - \mathbf{B}^x_{\mathbf{p}}(h_{\mathbf{p}} - h_{\mathbf{p}-\mathbf{e}_x})
$$
$$
+ \mathbf{B}^y_{\mathbf{p}+\mathbf{e}_y}(h_{\mathbf{p}+\mathbf{e}_y} - h_{\mathbf{p}}) - \mathbf{B}^y_{\mathbf{p}}(h_{\mathbf{p}} - h_{\mathbf{p}-\mathbf{e}_y}) \Big]
$$
$$
+ \mathcal{G}(h_{\mathbf{p}}) = \mathcal{F}_{\mathbf{p}}, \tag{B1}
$$

which can be rewritten as $\mathcal{H}(h_{\mathbf{p}}) = 0$, where

$$
\mathcal{H}(h_{\mathbf{p}}) = \alpha \mathbf{A_p} h_{\mathbf{p}}
$$
$$
+ \frac{\beta}{\Delta x^2} \Big[ \mathbf{B}^x_{\mathbf{p}+\mathbf{e}_x}(h_{\mathbf{p}+\mathbf{e}_x} - h_{\mathbf{p}}) - \mathbf{B}^x_{\mathbf{p}}(h_{\mathbf{p}} - h_{\mathbf{p}-\mathbf{e}_x})
$$
$$
+ \mathbf{B}^y_{\mathbf{p}+\mathbf{e}_y}(h_{\mathbf{p}+\mathbf{e}_y} - h_{\mathbf{p}}) - \mathbf{B}^y_{\mathbf{p}}(h_{\mathbf{p}} - h_{\mathbf{p}-\mathbf{e}_y}) \Big]
$$
$$
+ \mathcal{G}(h_{\mathbf{p}}) - \mathcal{F}_{\mathbf{p}}. \tag{B2}
$$

Eq. B2 can be solved by resorting to Newton's method (for a scalar):

$$
h_{\mathbf{p}} \leftarrow h_{\mathbf{p}} - \frac{\mathcal{H}(h_{\mathbf{p}})}{\mathcal{H}'(h_{\mathbf{p}})}, \tag{B3}
$$

where

$$
\mathcal{H}'(h_{\mathbf{p}}) = \alpha \mathbf{A_p} - \frac{\beta}{\Delta x^2}(\mathbf{B}^x_{\mathbf{p}+\mathbf{e}_x} + \mathbf{B}^x_{\mathbf{p}} + \mathbf{B}^y_{\mathbf{p}+\mathbf{e}_y} \mathbf{B}^y_{\mathbf{p}}) + \mathcal{G}'(h_{\mathbf{p}}). \tag{B4}
$$

## Appendix C:  A case for the treatment of the B coefficient in the FAS-MG algorithm

As mentioned in Sect. 3.1.3, the variable coefficient in the PDE Eq. 16 requires a special treatment due to coupling with the
main variable $h$. We tested two different approaches for the treatment of $\mathbf{B}$. In the first, which we will call *B fixed*, the value
of $\mathbf{B}$ is evaluated once per Picard iteration and frozen during the FAS solve. With the second approach, which we will call *B
on-the-fly*, $\mathbf{B}$ is recomputed and averaged down on each MG grid at the beginning of each V-cycle. We investigate the overall
efficiency of the algorithm in terms of number of Picard and total FAS iterations and CPU time as function of the FAS solve
tolerance. While keeping the (relative) tolerance of the outer Picard solver at a constant value of $10^{-8}$, statistics are collected
for about twenty time steps in a transient simulation (using the channelized test case described in Sect. 4.2) and the average
per time step is presented in Fig. C1. Across the entire range of FAS tolerances considered, the *B on-the-fly* is found to be
more efficient, with a lower average number of FAS iterations per step and smaller computational time (albeit by a smaller
margin due to the small computational overhead of recomputing $\mathbf{B}$ and updating the FAS multigrid hierarchy). When using the



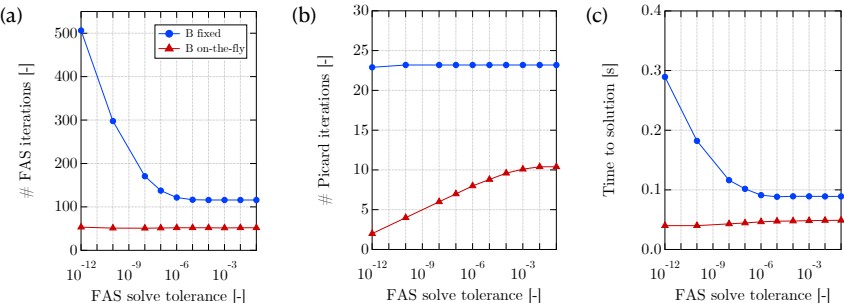

**Figure C1.** Average of a) number of FAS iterations, b) number of Picard iterations and c) total computational time required to complete one timestep with SUHMO, as function of the FAS solve tolerance. The tolerance of the Picard iterations is held fixed at $10^{-8}$.

.

*B fixed* approach, tight FAS tolerances lead to a large increase of the average FAS iterations count without having a significant effect on the Picard iterations count. In contrast, the *B on-the-fly* leads to a significant reduction of the Picard iterations count, because part of the nonlinearity is handled by the FAS.

Based on these results, we opt to use the *B on-the-fly* approach for all of our computations, and we fix the FAS tolerances to $10^{-10}$, appearing to provide a good compromise in terms of computational time.





**Appendix D: Details of runs $R_0$ to $R_4$**

| Name ($R_n$) | Number of levels | Base grid | Fine grid | $\Delta x^n$ |
|---|---|---|---|---|
| $R_0$ | 1 | 256x256 | — | 390.625 |
| $R_1$ | 2 | 256x256 | 512x512 | 195.3125 |
| $R_2$ | 3 | 256x256 | 1024x1024 | 97.65625 |
| $R_3$ | 4 | 256x256 | 2048x2048 | 48.828125 |
| $R_4$ | 5 | 256x256 | 4096x4096 | 24.4140625 |

**Table D1.** Details of the five AMR runs $R_n$.





*Author contributions.* AF and DM formulated the modeling approach. AF built the SUHMO model on a pre-existing framework from DM. AF performed simulations and compiled the paper with contributions from DM. EN provided overall guidance.

*Competing interests.* The authors declare that they have no conflict of interest.

*Acknowledgements.* Support for this work was provided through the Scientific Discovery through Advanced Computing (SciDAC) program
funded by the U.S. Department of Energy (DOE), Office of Science, Biological and Environmental Research and Advanced Scientific Computing Research programs, as a part of the ProSPect SciDAC Partnership. Work at Berkeley Lab was supported by the Director, Office of Science, of the U.S. Department of Energy under Contract No. DE-AC02-05CH11231.

We would like to thank Aleah Sommers and Colin Meyer for many fruitful discussions. AF would also like to thank Mathieu Morlighem and Basile de Fleurian for their help setting up test cases from the SHMIP intercomparison project.



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
