# Peer review of "SUHMO: an AMR SUbglacial Hydrology MOdel v1.0"

_Geoscientific Model Development, 2022_

## Referee Comment (RC1)

Review of "SUHMO: an AMR SUbglacial Hydrology MOdel v1.0"
Submitted by Felden et al. to *Geoscientific Model Development*

General comments:

This manuscript documents the development of a new model that simulates the evolution of water flow and pressure beneath glaciers and ice sheets. SUHMO adopts the general equations and continuum approach of the SHAKTI model (Sommers et al., 2018) with modifications to facilitate incorporation of adaptive mesh refinement (AMR), in order to resolve individual subglacial channels.

Overall, the paper is clearly written and well supported with references and figures. I do not find the simulations based on SHMIP tests to be especially helpful as there is no AMR and channel resolution involved in the SUHMO results presented. However, I understand the motivation of the authors to include those results as a sort of benchmark or verification of the approach, and I leave the decision up to them whether or not to retain those.

In general, some aspects of the numerical experiments and results presented in the paper could be more clearly explained. For example, the melt rate in SUHMO assumes no geothermal flux and no frictional heat. These can be large sources of basal melt, particularly frictional heat for fast-moving glaciers, and the decision to neglect these should be explained.

As already commented by the Editor in the online discussion: the model code, input, and output need to be made available in a repository that complies with GMD's guidelines.

Please see specific comments and technical corrections below.

With some revisions, I feel this manuscript should be published. As conveyed by the authors, challenges remain in incorporating the influence of subglacial drainage into ice dynamics models, and this AMR approach is a promising step to evaluate the role of individual drainage features and better understand the spatial resolution necessary to capture relevant subglacial pressure for large-scale ice sheet simulations.

Specific comments:

Line 11: Consider strengthening this last sentence by including the main point or points of what you find out about effective pressure as resolution increases (instead of simply saying you discuss it, which is vague).

Lines 72-73: Studies that have used subglacial hydrology modeling on real Greenland glacier domains find widespread areas of channelization (for example, Cook et al., 2020). In that light, I'm not sure that channels are only expected to occur in very localized areas – each channel is

distinct, yes, but with AMR you would need to have many areas of refinement that could cover a large area of a glacier bed. You may want to consider re-wording this sentence.

Line 73: You could define AMR here, in the first instance of using it.

Lines 74-75: Suggested re-phrasing: "We follow the approach of Sommers et al. (2018), with adaptations to implement a subglacial hydrology model using the Chombo AMR framework (Adams et al., 2001-2021)."

Line 75: Consider briefly describing what Chombo is here.

Line 102: What value of omega is used?

Line 118: The term in the melt rate that accounts for changes in the pressure melting point with changes in water pressure as written here is not negligible in places with steep topography, for example (see Creyts and Clarke, 2010 about supercooling). There are good reasons to drop this term, however, that have to do underlying assumptions about the ice and water pressure being equal at the interface.

Equation (6): Why introduce the general *b'* here and not in Eqn. (1)? I would recommend to either present a more thorough treatment of partially filled drainage, or just go with the assumption from the start that the gap being ice and bed is always filled with water.

Line 125: This method of using beta for the opening by sliding is based on GlaDS (Werder et al., 2013).

Line 130: Please describe the "creep length scale" and the physical justification for Equation (7). What qualitative behavior is captured with this equation? Why does creep shut off below a threshold? (Hint: it may have something to do with ice being supported by asperities on the bed). GlaDS and SHAKTI use the gap height (b) as this length scale. Why is that, and why should it be improved upon?

Line 131: Again, see comment above about b'.

Table 1: tau_b and G are both listed as 0. You should make clear in your results that the melt rate does not include contributions due to geothermal flux or frictional heat from sliding. Why?

Table 1: The description for omega is not very informative. You may want to include a brief description near Equation (2) about what this parameter is and how it functions to make the laminar-turbulent transition.

Table 1: Why did you choose this value of A that corresponds to very cold ice? The equations described assume that the ice and water are both at the pressure-melting-point temperature, so a larger flow law parameter would be more appropriate.

Line 151: The diffusion-like term is basically a diffusion of gap height. It may be worth explaining in more detail what this represents physically.

Equation (10): I would not consider the second term in the parentheses (the pressure-melting term) to be dissipation. I suggest naming it separately.

Eqn. (16): What is $\rho_c$?

Line 211: If only one Picard iteration is used, what's the point?

Equation (17) and Lines 214-216: I recommend using a different letter to indicate time in the superscript to avoid confusion with the flow law exponent n.

Line 231 and Eqn. (19): Here too, the use of little n as $n_{tot}$ could be confusing.

Line 237: The domain (64x16 m) used for the channelizing test case is small. What about boundary effects?

Line 239 – 30 m3/s is pretty high flow for a moulin!

Line 240: Neumann (flux) boundary condition at the outlet and Dirichlet (value) boundary condition upstream? I think this is probably a typo and should be switched. Typically, the Dirichlet condition would be at the outlet to set head equal to bed elevation (for example, equivalent to atmospheric pressure). And the no-flux Neumann condition makes sense for the upstream condition based on your results.

Line 242: I was confused on first reading this, thinking that the Gaussian was referring to a time-varying input rate, whereas it was already stated that the input rate is 30 m3/s. It would be helpful to include wording that makes clear that the Gaussian distribution refers to the spatial footprint of the point source at the bed.

Line 248: It may be worth commenting here about why this type of super-narrow, tall opening should not be interpreted as a physically realistic basal crevasse.

Line 302: What do you mean by the effective system? Is this a typo for "efficient", or some other intended meaning?

Lines 306-307: I thought the moulin input is specified as constant. It is not clear what is referred to as Gaussian here.

Figure 5b: How high does the discharge calculated by SHAKTI go in case A1? (The vertical axis does not extend sufficiently). This seems like an error somewhere (quite possibly in the SHAKTI results submitted to SHMIP).

Lines 337-343: 5180 m3/s into 63 moulins = 82 m3/s into each moulin. This seems very high and could benefit from better justification or explanation of why you choose to use extreme input. What happens with more realistic moulin inputs (not as high)? Do you produce channelization?

Figure 9 – suggest moving the x,y,z coordinate in panel a to the right lower side so that it is oriented where x=y=0 (the red dot locator helps orient, but I found it confusing nonetheless). "Total height" could be called "surface elevation" instead.
Figure 9 – panel b: is moulin input in m/s or m3/s?

Line 351: What specific criteria for gap height and melt rate were used to determine regridding?

Line 370: Great! This is a big deal and will help make the case for AMR. It would be helpful to include a number here of what the minimum resolution is that you find to resolve channelization.

Line 383: I am not sure that it is correct to say that melting of ice walls was discarded in the original derivation of the SHAKTI equations. The equations treat gap height as a one-dimensional quantity, so two-dimensional cross-sectional area of a channel was never part of that derivation.

Conclusions section: Make a point of quantifying the minimum mesh resolution that you find necessary for resolving channelized features somewhere in here as a main take-away point.

Lines 391-395: This paragraph is arguably true, but I suggest rewording to strengthen the case for resolving channels in SUHMO. Describe why this may be necessary and helpful in ice-sheet modeling. (The way it currently reads make it sound like coupling with BISICLES is motivated by making sure individual channels don't matter – but we don't know yet if that's the case until you do it).

Technical corrections:

Line 3: comma before "however"

Line 71: typo in "subglacial"

Line 82: convergence analyses… *are* presented (or convergence *analysis*… is presented)

Line 124: should be kg m$^{-3}$ (missing a negative sign in the exponent)

Line 162: Insert word "accuracy" after algorithm

Line 173: missing space after "of"

Figure 1. Should the schematics on the bottom left and right have delta $x_0$ (instead of delta x) in the grid level labels for consistency?

Line 276: Period after Figs.

Line 293: Extra ) after 2a

Line 308: Extra )

Figure 9: The orientation of the two panels being different is spatially confusing. Is it possible to orient them so the red dot location is more consistent between perspectives? For example, rotate panel b 90 degrees counter-clockwise. Similarly for Figure 10b.

Line 366: Missing period after Fig.

Line 366: Either includ gap height $b$, or remove $N$

Line 373: Using lower-case n is again slightly confusing here.

---

## Author Response (AR1)

The authors would like to thank all of the reviewers for their helpful comments and suggestions. We have done our best to respond and we believe we have addressed all major concerns.

This letter is structured as follows: answers to the main questions/concerns raised by both reviewers are given first while minor comments are discussed in a second section. Typos ("technical corrections" from reviewer #1 and "line by line" from reviewer #2) are simply corrected in the revised manuscript. When relevant, references to specific reviewer's comments are provided in **italic bold.**

We hope that this version will be suited for publication to GMD.

I.	*General comments:*

** Reviewer #1:

*"...For example, the melt rate in SUHMO assumes no geothermal flux and no frictional heat. These can be large sources of basal melt, particularly frictional heat for fast-moving glaciers, and the decision to neglect these should be explained."*

Thank you for catching this! There was a mistake in Table 1. The typical value for the Geothermal flux is 0.05 W/m2, while the expression for tau follows a modified Weertman type: $\tau = c_b N \cdot u_b^{-2}$, where $c_b = 20^2$ (this value of $c_b$ has been tuned to reproduce pressure and discharge results in all of the different SHMIP test cases). Table 1 has been rectified.

Note however that both of these components of the melting rate (the geothermal flux and frictional heat) CAN be turned off if desired via keywords in the inputs file. To reach proper temporal convergence (and to produce the desired discharge) in SHMIP test case A, the geothermal flux has indeed been turned off - as was done in SHAKTI's contributions to SHMIP. Activating it increases the discharge, proportionally more for the test cases with the least recharge (A1-A3), but it does not affect the averaged effective pressure results significantly.

*"... the model code, input, and output need to be made available in a repository that complies with GMD's guidelines."*

The authors apologize for failing to comply fully with these requirements at the time of submission. We have registered both SUHMO v1.0 (https://doi.org/10.5281/zenodo.7045001) and the forked version of Chombo v3.2 ( https://doi.org/10.5281/zenodo.7040610 ) on Zenodo, including scripts/inputs and post-processing tools enabling any user to reproduce all results presented in the paper.

** Reviewer #2:

*"... I would prefer to see SHMIP experiments where there is significant channelisation. Note that the SHMIP instructions suggest to tune to the GlaDS output of runs A3 and A5, with the latter being channelised up to mid-domain in GlaDS. …"*

The instructions provided on the SHMIP paper/website to (re)produce the results of the different suites come with a list of parameters, for which standard values are provided. It is our understanding that these values should be preferred, and that the tuning should only adjust those parameters specific to each model. In the case of SUHMO (and also SHAKTI) the list of such parameters is, in reality, quite short: the turbulent parameter omega, the basal shear stress and the geothermal flux are the only candidates. The opportunities for tuning our model while still matching the proper discharge (not generating too much basal melt) are therefore limited.

Additionally, note that GlaDS (as many other channelizing models included in SHMIP) uses two different equations to solve separately for sheets and channels. Effectively, this would mean that to fully tune to such a model, one would have to artificially shut off the opening by sliding in Eq. 6 in some areas of the domain to see the nucleation of a channel. This can be done, and we have experimented with adding some roughness to the $b_r$ term -the typical height of bed bumps. Results are reported hereafter, for test case A5. Note that this does enable channelization; however, we feel that justifying such a profile for $b_r$ would be difficult and would not bring much to the discussion - since our ultimate goal was to demonstrate that our implementation of the equations (shared by SHAKTI) was done properly.

[Figure]

As one can see, results visually resemble those of GLaDS (a better agreement could probably be reached if more time was dedicated to tuning and if more parameters were allowed to be tuned), in that the inception of a couple of channels is seen. This at least confirms that perturbations and irregularities (such as those encountered when using an unstructured mesh) can lead to different flow configurations.

*"... the shown suite-B is channelised in GlaDS but not in SUHMO & SHAKTI."*

The previous comments about the fundamental model differences and the need to comply with the SHMIP provided parameters values apply here as well. One could think that channel nucleation could simply be driven by the external source terms in this case, but channelization can only be generated with the set of parameters provided if randomness is added to $b_r$ , effectively preventing opening by sliding locally.

*"... showing a comparison to GlaDS outputs for SHMIP cases would be informative, as that is currently the canonical distributed+channelised drainage model."*

Leaving aside the question of which model is canonical, we are primarily using our SHMIP runs to demonstrate a verification of our model, and so comparisons with SHAKTI are most appropriate in that regard since it is the closest in formulation to SUHMO. In doing so and tuning to the same parameters -and showing that we exhibit the same behavior, we expected for our model to be "verified by the transitive property" (bearing some resemblance with how model "og" was verified by reproducing "mw" results in SHMIP). Our ultimate goal is to be able to use SUHMO to tackle problems such as very big domains with very refined structures - where the power of AMR would become useful. The small problems provided in

SHMIP do not really enable us to exercise these features and so we chose to use them solely to provide a verification of our implementation.

*"SHMIP suite E…it would be nice and interesting to see results of SUHMO results with pressure-melt term turned on."*

The authors agree with the reviewer that suite E was originally designed to have the pressure-melt turned on and that it would have been preferable to see results for SUHMO/SHAKTI with that term activated. It is possible to activate this term in SUHMO, however, we have found several problems with this test case. The most important one is that values for the maximum water-filled gap height do not seem to be converging, even if the simulation is run for 50+ years, no matter the choice of parameters. Another issue concerns the description of the test case. Indeed, reported values for the channelization (in the supplementary material of the SHMIP paper) are difficult to interpret (values for the discharge are provided as "from" and "more than" for example), which made it difficult to analyze our results properly. Finally (see example below), while we do retrieve similar tendencies as "mw", overall effective pressure values are quantitatively different, especially in the right part of the domain (in x). Here also, we found that it would be potentially distracting from the main goal of the manuscript to justify our choices for parameter tuning and discuss all of these discrepancies. Thus, we decided to focus on validating our implementation, which relies on a structured mesh (for which such geometries are not straightforward), on the case without the pressure term and compare ourselves with, e.g., model "og'", and all the other models that do not include it.

[Figure]

As an example, two representative E test cases are reported above: E1 and E4, where the omega has been increased from the nominal value of 0.001 to 0.005, and the Cb value has been reduced to 10. We see that, as the overdeepening increases (from E1 to E4), ice creep increases also in the portion of the bed exhibiting negative gradients of bed elevation (in the middle of the domain near the outlet), which is the expected behavior. This is accompanied by a switch from channel-like flow configuration to a sheet-like flow configuration. Likewise, the effective pressure profiles follow more or less the same tendencies -albeit with different overall values- as those reported for the "mw"/"og" results. We do see significantly different behavior far from the outlet (in the right half of the domain in

x), which are clearly underlined in the plots of effective/ineffective discharge. These results and analysis has been included as a Supplementary Material.

*"The authors mention in Section 6 that for channelisation to occur resolution needs to be good enough. It would be good to elaborate on this point some more. Does the used refinement criteria allow to always capture the channels or could some be un-noted and thus not simulated?"*

We have reworded wherever possible to make it clear that we do not expect for the case in Sect. 6 to be sufficiently resolved to be spatially converged, even with a total of 5 levels. We have added a paragraph to clarify that this is only a qualitative analysis, requiring further investigations and a better quantitative analysis on a real test case (this is ongoing work which we plan to present in a follow-on paper).

The reviewer raises a significant point here concerning refinement criteria. In SUHMO, two variables are usually employed as refinement criteria: the melting rate and the water-filled gap height. Ideally in AMR simulations, a feature is first captured on the coarsest mesh, and its representation improved on subsequent levels. It follows that if the first level is not so coarse as to miss it entirely, the feature will end up being "properly resolved" (assuming the user has allowed enough refinement levels to do so). The reviewer is correct that it is thus possible for very fine structures to be missed entirely if the criteria (the melt rate, for example) ends up being inadequate to catch them. However, our experience with AMR and the SHAKTI formulation is that the model is able to predict the broad onset of channelization, and then we observe that very refined structures (channels) usually stem from refining these coarser ones, which come from even coarser ones etc. So, what usually happens is that small structures end up being captured/revealed by the subsequent refinement of coarser ones, with the corresponding increase in solution quality and accuracy.

*"Also, in the small-scale experiment (Fig. 4) the channel is resolved with ca. 10 nodes across its width. How well resolved is the channel in Fig.11 for R4? Should a channel always be resolved with several grid points across its width? Or is one enough? I think it would be ok to leave these questions open for a future study but mentioned they should be in the discussion"*

The channel in Fig. 11 with the finest resolution can only be resolved to $\Delta x$, which is ~25 meters. We do not believe this resolution to be sufficient to capture all flow features in this example. As can be seen in the middle panel (delta N) of Fig. 11: the effective pressure distribution is still evolving with refinement, indicating remaining under-resolved flow features. Interestingly, an investigation of the main valley 1 channel reveals that the channel can sometimes have as many as 4 points in its width. As previously emphasized, this test case is purely synthetic – it appears that the produced channel could be significantly larger in width than found or expected in real life situations. We have amended the text to make it clearer that our intent was not to demonstrate perfect convergence for the AMR test case of Sect. 6.

To determine if a channel must be resolved with more than one point across its width for its impact to be accurately represented is an open question we plan to address in future work, and a sentence has been added to emphasize so. Likewise, what resolution is effectively sufficient in the context of providing accurate boundary conditions to an ice-sheet model also remains an open question for future exploration.

*"It would be interesting, and I think eventually necessary, to compare the SUHMO simulated channels to more classically simulated R-channels such as in GlaDS. Do they conduct the same discharge at the same pressure gradient? How do parameters*

*translate between a classic R-channel formulation and this regularised sheet formulation?"*

The authors completely agree with the reviewer on that point. We consider that to be beyond the scope of this model description paper, and it will be the subject of future work.

II.   *Minor concerns:*

** Reviewer #1:

*Line 11: Consider strengthening this last sentence by including the main point or points of what you find out about effective pressure as resolution increases (instead of simply saying you discuss it, which is vague).*

The last part of the Abstract has been modified to emphasize our results/ perspectives.

*Line 75: Consider briefly describing what Chombo is here.*

A sentence has been added.

*Line 102: What value of omega is used?*

The typical value for omega is 0.001, as provided in Table 1.

*Line 118: The term in the melt rate that accounts for changes in the pressure melting point with changes in water pressure as written here is not negligible in places with steep topography, for example (see Creyts and Clarke, 2010 about supercooling). There are good reasons to drop this term, however, that have to do underlying assumptions about the ice and water pressure being equal at the interface.*

We have modified the text to underline the fact that this Pw term is indeed sometimes relevant. Note that we do include it in the big test case of Section 6, as well as in our convergence analysis studies. We did drop it from the SHMIP test cases which use the parabolic ice profile.

*Equation (6): Why introduce the general b' here and not in Eqn. (1)? I would recommend to either present a more thorough treatment of partially filled drainage, or just go with the assumption from the start that the gap being ice and bed is always filled with water.  (*see also *Line 131: Again, see comment above about b'.)*

This has been addressed by removing the variable b' altogether and starting the derivation from the assumption that the drainage space is always filled – *Note that Reviewer#2 had the same suggestion.*

*Line 125: This method of using beta for the opening by sliding is based on GlaDS (Werder et al., 2013). (*see also *Table 1: The description for omega is not very informative. You may want to include a brief description near Equation (2) about what this parameter is and how it functions to make the laminar-turbulent transition.* and reviewer #2 comment: *about 110: it would be nice to contrast this formulation to the more commonly used Manning or Darcy-Weisbach formulation. )*

We have added the reference for beta.

Our intent with Sect. 2.1 was to be brief with the summary of the set of equations –since we are using the same basic set of equations as that of Sommers et al. 2018 - and then to describe in detail the unique numerical aspects of our implementation. A sentence has been added at the beginning of Sect. 2.1 to emphasize this and refer the reader to the SHAKTI model description paper, where the authors cite all relevant work and explain/discuss the physics behind the formulation of the set of equations in much more depth than the present manuscript (see for instance the discussion following eqs. 6-8 in Sommers et al., 2018 for a very thorough explanation of how the momentum equation compares to other formulations).

***Line 130: Please describe the "creep length scale" and the physical justification for Equation (7). What qualitative behavior is captured with this equation? Why does creep shut off below a threshold? (Hint: it may have something to do with ice being supported by asperities on the bed). GlaDS and SHAKTI use the gap height (b) as this length scale. Why is that, and why should it be improved upon?***

Eq. 7 has been augmented with an explicative paragraph. Indeed, this formulation is different from what was employed in SHAKTI. This formulation has been introduced to have more control over the closure of sheets, such as when these are known to exist, for example – *Note that Reviewer#2 had the same suggestion.*

***Table 1: Why did you choose this value of A that corresponds to very cold ice? The equations described assume that the ice and water are both at the pressure-melting-point temperature, so a larger flow law parameter would be more appropriate.***

The authors chose to follow the recommendations provided in the SHMIP paper, for which this specific value of A was specified. As all other simulations performed and reported in this paper were not intended to be an exact representation of reality but more a validation of the implementation/numerical framework, we did not seek to change that value throughout this paper.

***Line 151: The diffusion-like term is basically a diffusion of gap height. It may be worth explaining in more detail what this represents physically.***

While we do think of this term as a diffusion-like term -especially in the way we treat it numerically, we want to be careful in calling it that directly, mainly because when attempting to derive it theoretically from the constitutive equations, it ends up looking more like a convective term (the dimensional analysis backs this up). A precise, mathematically-sound justification for this term is still ongoing work.

***Line 211: If only one Picard iteration is used, what's the point?***

As can be seen from Eq.16, the RHS still depends on h in a non-trivial way. Picard iterations are necessary to treat these remaining non-linearities. While the required number of Picard iterations is usually –and on average, very low, there are some instances, particularly during initializations or transient channel formations, where this number increases significantly (never for long). We have added a sentence clarifying this.

***Line 237: The domain (64x16 m) used for the channelizing test case is small. What about boundary effects? + Line 239 – 30 m3/s is pretty high flow for a moulin!***

We do not expect for boundary effects to matter, but even if they did we would not be too worried about it as the intent with this test case is to provide an easy, quick, channelizing test to perform convergence studies. Likewise, the moulin inflow is not intended to be physically meaningful, but merely to provide for an easily channelizing test case.

***Line 240: Neumann (flux) boundary condition at the outlet and Dirichlet (value) boundary condition upstream? I think this is probably a typo and should be switched. Typically, the Dirichlet condition would be at the outlet to set head equal to bed elevation (for example, equivalent to atmospheric pressure). And the no-flux Neumann condition makes sense for the upstream condition based on your results.***

This is correct. Thank you for catching this. It has been rectified.

***Line 242: I was confused on first reading this, thinking that the Gaussian was referring to a time-varying input rate, whereas it was already stated that the input rate is 30 m3/s. It would be helpful to include wording that makes clear that the Gaussian distribution refers to the spatial footprint of the point source at the bed. (*** see also:

***Lines 306-307: I thought the moulin input is specified as constant. It is not clear what is referred to as Gaussian here.)***

The authors hopefully clarified the description of the moulin setup.

***Figure 5b: How high does the discharge calculated by SHAKTI go in case A1? (The vertical axis does not extend sufficiently). This seems like an error somewhere (quite possibly in the SHAKTI results submitted to SHMIP).***

The authors used the same set of data as that reported in the paper and supplementary material by De Fleurian et al. 2018 (the SHMIP paper). Case A1 does exhibit a very large discharge at the domain's outlet.

***Lines 337-343: 5180 m3/s into 63 moulins = 82 m3/s into each moulin. This seems very high and could benefit from better justification or explanation of why you choose to use extreme input. What happens with more realistic moulin inputs (not as high)? Do you produce channelization?***

It is specified in the introduction of Section 6 that *"The water delivered by the moulins is constant, no seasonal cycle is considered, and is also quite high: this experiment should be taken as a demonstration of the robust behavior of the system even under prolonged high melting scenarios, when a high degree of channelization is expected. Our purpose is to demonstrate the importance of spatial resolution when looking at subglacial water patterns, and ultimately to examine the impact of resolution on the effective pressure distribution."* Indeed, this test case is purely synthetic and it looks as if the produced channel could be significantly larger in width than found or expected in real life situations. While the use of unrealistic water input (or even topography) may look a bit odd to a seasoned glaciologist, we do not believe that this invalidates the conclusions drawn here about the need for a minimal resolution to properly resolve (or at the very least capture) flow features.

That being said, we absolutely agree with the reviewer that SUHMO needs to be applied to real, non steady-state, scenarios with realistic topographies and moulin inputs/locations. This is a significantly larger endeavor, however, and we decided to limit ourselves to mathematical/algorithmic considerations in this paper. Applying SUHMO to a realistic glacier/drainage basin where we can also draw conclusions about the physics of the phenomenon at play is very exciting, and is ongoing work.

***Figure 9 – panel b: is moulin input in m/s or m3/s?***

In accordance with Eq. 11, the moulin input is provided in m/s.

***Line 351: What specific criteria for gap height and melt rate were used to determine regridding?***

The criteria used for the melt rate is a minimum of 2e-5 kg/m2/s - all cells with a melt rate higher than that are automatically tagged for refinement. The criteria used for the gap height is a minimum of 0.1 (but this is effectively only used for R4, since for R3 and under, the gap height is smaller). We have added a sentence detailing this.

***Line 370: Great! This is a big deal and will help make the case for AMR. It would be helpful to include a number here of what the minimum resolution is that you find to resolve channelization.*** (see also: ***Conclusions section: Make a point of quantifying the minimum mesh resolution that you find necessary for resolving channelized features somewhere in here as a main take-away point.***)

We have added an entire paragraph to build on our results and discuss the resolution requirements (as well as underline the need for further investigations), in Sect. 6.3.

***Lines 391-395: This paragraph is arguably true, but I suggest rewording to strengthen the case for resolving channels in SUHMO. Describe why this may be necessary and helpful in ice-sheet modeling. (The way it currently reads make it***

*sound like coupling with BISICLES is motivated by making sure individual channels don't matter – but we don't know yet if that's the case until you do it).*
    We have expanded the conclusion section to more clearly make this case.

\*\* Reviewer #2:
    *Please only use abbreviations if really needed, i.e. reducing the length considerably and reducing line-noise for the reader. Thus remove the abbreviations "GrIS", "AIS", "GMSL", "GC". I'm a bit unsure about the "AMR" in the title, albeit it should be ok for GMD.*
    GrIS and AIS have both been removed. The authors feel like the use of mathematical abbreviations such as GC (for ghost-cells) and AMR (for Adaptive Mesh Refinement) should be allowed, as they are common and employed throughout the document (AMR is used a total of 46 times in the manuscript). Likewise, GMSL is employed a total of 8 times and using this commonly employed abbreviation reduces unnecessary repetitions.
    *One of the main aims of this model seems to be eventually coupled to the BISICLES ice sheet model. I think this is worth mentioning in the abstract.*
    This has been addressed.
    *One of the main reasons that subglacial hydro modeling is not further along is the lack of observations. This should be mentioned in the Abstract and Introduction.*
    A sentence was added in the Introduction reflecting this point.
    *It would be good to look at convergence also for the case of section 6 as that features a channel.*
    The authors have adapted the text throughout the document to make it clear(er) that we do not expect for the AMR test case of section 6 to reach spatial convergence, even with a total of 5 AMR levels. The goal here was to bring forth tendencies and to provide a qualitative more than quantitative analysis of subglacial flow features as resolution increases - basically, to show that there is a need to dig deeper and that SUHMO is the tool to help with that. Further (ongoing) work will focus on conducting a more thorough refinement study analysis on relevant topographies with realistic water input.
    *I feel that the SHAKTI parameters used in SHMIP lead to too much sheet flow and not enough channel flow compared to the other models in SHMIP. I don't know what the reason was to pick those parameters, maybe SHAKTI struggles with high channelisation or the parameters were just the optimal ones. Irrespective of the reason, the SUHMO-results presented, based on the SHMIP-SHAKTI parameters, end up having to use very high inputs (both in run for Fig 4 and Fig 11) to get channels.*
    See previous comments in the "General comments" section concerning the tuning of parameters for SUHMO.
    Note that the run of Fig. 4 was not specifically tuned to run with realistic moulin input. This test case was chosen to provide for a small, quick and channelizing test case to challenge our numerical implementation and demonstrate second-order spatial convergence of our algorithm.
    *It would be good to have a master-script (or master make-file) which produces all the figures in this manuscript. Maybe there is one already, but 2min of browsing the github repo did not turn up one. State this in the "code availability" section.*
    The authors apologize for failing to comply with these instructions at the time of submission. We have registered both SUHMO v1.0 (https://doi.org/10.5281/zenodo.7045001) and the forked version of Chombo v3.2 (

https://doi.org/10.5281/zenodo.7040610 ) on Zenodo, including scripts/inputs and post-processing tools enabling to reproduce all results presented in the paper.

*III. Note about typos:*
** Reviewer #1:
   **Figure 1. Should the schematics on the bottom left and right have delta x0 (instead of delta x) in the grid level labels for consistency?**
      The images are intended to be representative of any pair of levels in the V-cycle, with generic grid size $\Delta x$.
** Reviewer #2:
   **9: should it read "MultiGrid" to match abbreviation?**
      MG is the universally accepted abbreviation, but they are generally referred to as "multigrid" methods
   **20: that "likely" is probably IPCC speak, why not reference the latest IPCC report?**
      The reference is provided at the end of the sentence (Masson-Delmotte et al., 2021)
   **29: also cite Gilbert et al 2022 https://doi.org/10.1029/2021GL097507**
      It sounds like this paper "breaks with the common view on the subglacial environment that water pressure drives the sliding speed by modulating friction at the glacier base." which is unfortunately what we are trying to say in this sentence (although this work is very interesting, and we thank the reviewer for pointing it out!).
   **67: write "The SHAKIT model..." and 67: isn't it "SHaKIT"?**
      The model by Sommers and coworkers is commonly referred to as the SHAKTI model (for "Subglacial Hydrology and Kinetic, Transient Interactions", see also https://issm.jpl.nasa.gov/documentation/hydrologyshakti/).
   **This reviewer also found several figures to be too small (1, 2, 9, 10, 11) and suggested that Fig. 3 should not present both the head and Pw.**
      Figs. 2 and 9 were chosen by the authors to be in the half column format. All other figures are full width. We argue that Fig. 2 and 9 should remain in the half column format, however, we have increased the font sizes for all of these figures and reworked them to make small features bigger. We believe them to be legible in the current format.
      We have added two scales to Fig. 11 to help the reader's eye and augmented the text in the legend of Fig. 2 to emphasize the change of scale for (c). Note that the channel width, in the most refined R4 case, varies spatially from one to 4-5 cells (about a km). As such, we do not believe the channels to be converged. We have attempted to clarify this fact throughout the manuscript.
      Pw indeed follows the same trend as the head in Fig. 3. We chose to add it to account for the evolution of N, the effective pressure - since b,h and N are the 3 main variables. We changed Fig. 3 to add N instead of Pw.
   **The reviewer suggested to cite many fundamental papers in the introduction as well as in the description of our model's equation**
      We have accommodated most of them. We have also referred the reader to the SHAKTI paper by Sommers et al. 2018 in Section 2.1, as our intent here was to be brief in describing the set of equations (that we simply reuse).

---

## Author Response (AR2)

A summary of all modifications to the final revised version is provided hereafter. Thank you for taking the time to review our revised manuscript.

**# General comments of Reviewer 1 (R1) & R2**
**R1 requests availability of code and data in a form suitable for GMD.**
**- The code is now available in a suitable format, including all scripts to run the presented simulations. However, the Zenodo links need to be added to the "Coda availability" section.**
**- However, there is no model output data provided. I'm not sure whether this is a requirement; even if it is not required, model run data would be useful to have. Most useful would probably be the data format used in the SHMIP exercise.**

A new version of SUHMO has been tagged and referenced in the manuscript. This new version includes the following additional input files/scripts/results:
- inputs and scripts to run the distributed test cases of Sect. 4
- convergence analysis results for both distributed and channelized test cases presented in Sect. 4
- results for all one-dimensional plots presented in Sect. 5

Note that each test case is structured as follows: a main folder (say, SHMIP_A for the A suite) contains subfolders for each specific run (say, A1 to A6) where input files are provided for the main run and (when applicable) to launch a series of post-processing. Alongside these run subfolders, a (or a series of) python script(s) is (are) present to semi-automatically launch all of the runs and their analysis, and produce all data reported in the manuscript.

For convenience and easy testing of reproducibility, all of the 1D results presented in the paper are provided in plain text (.dat, to be as general as possible). These data files are located in "results" subfolders under the appropriate run subfolders (for SHMIP) or in a CONV_ANA subfolder at the root of the main folder (for results of Sect. 4).

Including the entirety of the 2D results presented in the paper ("plotfiles") would take too much storage space, especially for the 2D fields of Sect. 6. The aforementioned python scripts will produce these 2D results, however (for all runs of Sect 4, 5 and 6 along with the modified tests of suite E as presented in the Supl. Mat.). These plotfiles will be in HDF5 format. These HDF files can be visualized with several open source software -e.g., ParaView. This has all been precised in the "Code Availability" Section.

**R2 was wondering about tuning SUHMO to the GlaDS results in SHMIP:**
**- it's fine to not do that but:**
**- state clearly that SHMIP runs are mostly done for a 1-1 comparison to SHAKTI and not so much to for a comparison to other models. Add this to the first paragraph of Section 5.**
This has been addressed in Sect. 5.

**# Specific comments of R1**
**Re-Remark to R1, Table 1, value of A: the SHMIP paper recommends to use A=3.35e-24 Pa^3s^-1 (which is reasonable for temperate ice), however that is for a closure formulation of form 2A/n^n S N^n. Here a form A' S N^n is used and thus the used A'=2.5e-25 corresponds to the recommended A=3.35e-24 Pa^3s^-1.**
This has been specified in the legend of Table 1.

**# Specific comments of R2**
**Hewitt 2011 needs to be cited https://doi.org/10.3189/002214311796405951. This is the base for this model and thus the original warrants a citation even though the specifics are found in SHAKTI. Probably best in the first paragraph of Section 2.**

This has been rectified.